# Peroxisomal targeting of a protein phosphatase type 2C via mitochondrial transit

Thorsten Stehlik[1], Marco Kremp[1], Jörg Kahnt[2], Michael Bölker[1,3 ✉] & Johannes Freitag[1 ✉]

Correct intracellular distribution of proteins is critical for the function of eukaryotic cells. Certain proteins are targeted to more than one cellular compartment, e.g. to mitochondria and peroxisomes. The protein phosphatase Ptc5 from *Saccharomyces cerevisiae* contains an N-terminal mitochondrial presequence followed by a transmembrane domain, and has been detected in the mitochondrial intermembrane space. Here we show mitochondrial transit of Ptc5 to peroxisomes. Translocation of Ptc5 to peroxisomes depended both on the C-terminal peroxisomal targeting signal (PTS1) and N-terminal cleavage by the mitochondrial inner membrane peptidase complex. Indirect targeting of Ptc5 to peroxisomes prevented deleterious effects of its phosphatase activity in the cytosol. Sorting of Ptc5 involves simultaneous interaction with import machineries of both organelles. We identify additional mitochondrial proteins with PTS1, which localize in both organelles and can increase their physical association. Thus, a tug-of-war-like mechanism can influence the interaction and communication of two cellular compartments.

[1] Department of Biology, Philipps University Marburg, Marburg, Germany. [2] Max Planck Institute for Terrestrial Microbiology, Marburg, Germany. [3] LOEWE Center for Synthetic Microbiology, Marburg, Germany. ✉email: boelker@staff.uni-marburg.de; johannes.freitag@biologie.uni-marburg.de

Peroxisomes are dynamic and versatile single membrane organelles that play an essential role in the breakdown of fatty acids and the detoxification of hydrogen peroxide[1,2]. Different pathways have been characterized, which mediate import of fully folded and even oligomeric proteins from the cytosol into the peroxisomal matrix[1,3]. The majority of peroxisomal matrix proteins contains a C-terminal extension known as peroxisomal targeting signal type 1 (PTS1), which is recognized by the cytosolic receptor Pex5[4,5]. PTS1 motifs have the prototype sequence S-K-L, but variations of this motif exist and the sequence context of the tripeptide influences targeting efficiency[4,6]. Cargo-bound Pex5 docks at a complex containing the peroxisomal membrane proteins Pex13 and Pex14. The receptor–cargo complex integrates into the peroxisomal membrane and releases its cargo into the peroxisomal matrix. Recycling of Pex5 depends on ubiquitination and the heterohexameric AAA-ATPases Pex1/Pex6[3,5,7]. Several proteins are imported into more than one cellular compartment. A variety of strategies has evolved to achieve dual or multiple targeting, e.g., gene duplication, differential splicing, and alternative protein translation[8–13]. Here, we report an alternative mechanism for dual targeting. The protein phosphatase Ptc5 is sorted to mitochondria and processed in the inner mitochondrial membrane before translocation to the peroxisomal matrix. This mechanism resembles a tug-of-war-like scenario and can influence the association of mitochondria and peroxisomes. We identify further proteins likely following a similar pathway.

## Results

**Dual targeting of Ptc5 to mitochondria and peroxisomes**. In a computational survey of *Saccharomyces cerevisiae* proteins containing a PTS1, we detected a protein with unexpected domain architecture. Although PTS1 is usually used as a signal for the import of soluble proteins into peroxisomes[1], the type 2 C protein phosphatase Ptc5 contains both a functional PTS[14] and a transmembrane domain. In addition, Ptc5 harbors an N-terminal mitochondrial presequence (Fig. 1a). Ptc5 has been previously suggested to dephosphorylate mitochondrial pyruvate dehydrogenase, which was later put into question by proteomics data[15,16]. During import into mitochondria Ptc5 is processed by the mitochondrial inner membrane peptidase (IMP) complex and released into the intermembrane space[17–19]. The specific domain architecture of Ptc5 is conserved in other fungi, suggesting biological relevance (Fig. 1b). We analyzed the intracellular localization of Ptc5 by expression of a C-terminal tagRFP fusion extended by the PTS1 of Ptc5 (Ptc5-RFP-PTS) to preserve both targeting signals. The *tef1* promoter was used for expression of all tagRFP fusion proteins throughout the study. Ptc5-RFP-PTS was expressed in strains either containing the mitochondrial inner membrane protein Tim50 fused to YFP, or the peroxisomal ATP transport protein Ant1 fused to YFP. Respective genes were tagged at the endogenous locus[20]. Ptc5-RFP-PTS localized in mitochondria and in peroxisomes (Fig. 1c, Supplementary Fig. 1a, b). A control protein without PTS1 (Ptc5-RFP) was only detected in mitochondria. Peroxisomal targeting of Ptc5-RFP-PTS was not observed in Δ*pex5* cells and in other mutants defective in peroxisomal import (Fig. 1d, Supplementary Fig. 1c). Physical interaction of the PTS1 of Ptc5 with Pex5 was demonstrated with a yeast two-hybrid assay[21] (Supplementary Fig. 1d). Dual targeting of Ptc5-RFP-PTS to mitochondria and peroxisomes was confirmed by density gradient centrifugation. Ptc5-RFP-PTS comigrated both with the mitochondrial outer membrane protein Por1 and the peroxisomal matrix protein GFP-Sps19, whereas Ptc5-RFP was predominantly detected in mitochondrial fractions (Fig. 1e, Supplementary Fig. 1e).

Next, we followed the localization of an endogenously expressed and internally Myc-tagged Ptc5 (Ptc5-3xMyc-PTS) by immunofluorescence microscopy and density gradient centrifugation (Fig. 2). Dual targeting of Ptc5-3xMyc-PTS was detected with both experimental strategies (Fig. 2). A smaller fraction of Ptc5-3xMyc-PTS (~10%) compared to Ptc5-RFP-PTS (~30%) colocalized with peroxisomes (Fig. 2b). A similar quantitative difference was obtained when results from density gradient experiments were compared (Fig. 2d). This may result from stable folding of tagRFP prior to complete import into mitochondria. Recent proteomics data indicates a physical association of Ptc5 with the peroxisomal membrane protein Pex14 (ref. [22]). Thus, in addition to its previously reported localization in the mitochondrial intermembrane space, Ptc5 is also targeted to peroxisomes.

**Ptc5 processing in mitochondria prior to peroxisomal import**. To our surprise, deletion of *imp1*, encoding one of the catalytic subunit of the mitochondrial IMP complex, interfered with the sorting of Ptc5-RFP-PTS to peroxisomes (Fig. 3a, Supplementary Fig. 2). Analysis of protein fractions derived from wild-type (WT) cells by high-resolution SDS gel electrophoresis and western blotting revealed identical electrophoretic mobility of both mitochondrial and peroxisomal Ptc5-RFP-PTS. Unprocessed Ptc5-RFP-PTS isolated from *imp1* mutant cells migrated with reduced mobility (Fig. 3b). Carbonate extraction experiments confirmed removal of the N-terminal transmembrane domain of mitochondrial and peroxisomal Ptc5-RFP-PTS by Imp1 (Supplementary Fig. 3). Deletion of *imp2*, which encodes the second catalytic subunit of the IMP complex[18], also decreased peroxisomal localization of Ptc5-RFP-PTS (Fig. 3c). To test if targeting of Ptc5 to peroxisomes is impaired by the mitochondrial respiratory defect of *imp* mutants[19], we analyzed the localization of Ptc5-RFP-PTS in Δ*coq9* cells. Coq9 is required for biosynthesis of ubiquinone and respiratory growth[23]. Sorting to peroxisomes was not compromised in this mutant (Fig. 3c). We also observed that the deletion of *imp1* interferes with processing of endogenously tagged Ptc5-3xMyc-PTS (Fig. 3d). These data demonstrate that processing by the mitochondrial IMP complex is not only required for release into the intermembrane space, but also critical for peroxisomal targeting of Ptc5.

**Characterization of the IMP cleavage site in Ptc5**. To improve discrimination between unprocessed and mature forms of RFP-tagged Ptc5 on SDS–PAGE, we constructed C-terminally truncated Ptc5 variants (Ptc5$^{1–201}$-RFP-HA and Ptc5$^{1–201}$-RFP-PTS). Both versions behaved like full-length proteins and displayed identical electrophoretic mobility patterns independent of their targeting signal (Fig. 3e, Supplementary Fig. 4a, b). To provide complementary evidence that peroxisomal import of Ptc5 requires processing by Imp1, we sought to delete its cleavage site in Ptc5. We affinity purified Ptc5$^{1–201}$-RFP-HA from WT and Δ*imp1* cells, and mapped the cleavage site for Imp1 between amino acid position 83 and 84 by liquid chromatography–mass spectrometry (LC–MS; Fig. 3f, Supplementary Data 1). These data were confirmed by a series of truncations and a point mutation changing the amino acid sequence around the cleavage site (Supplementary Fig. 4c, d). Deletion of the amino acid stretch from position 81 to 84 (LSLD) completely abolished cleavage and interfered with peroxisomal localization (Fig. 3g, h). To exclude a peroxisomal fraction of the IMP complex, we analyzed the localization of endogenously tagged and biologically active Imp2-3xMyc by fractionation experiments (Fig. 4). Imp2-3xMyc was only detected in mitochondrial fractions (Fig. 4a). Accordingly, a Ptc5-RFP-PTS variant lacking the mitochondrial presequence but retaining the transmembrane domain and the Imp1 cleavage site

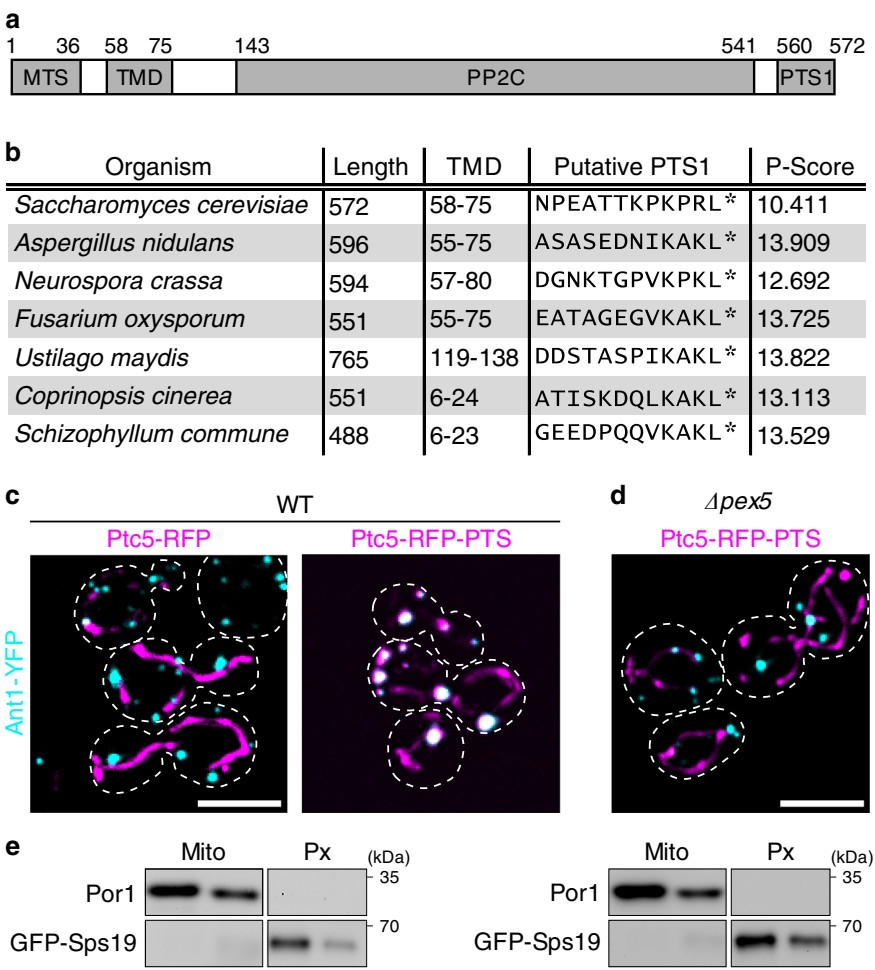

**Fig. 1 Ptc5 is localized in mitochondria and in peroxisomes. a** Domain structure of Ptc5 from *S. cerevisiae*. MTS, TMD, and PTS1 refer to the mitochondrial targeting signal, the transmembrane domain and the peroxisomal targeting signal, respectively. PP2C specifies the position of the phosphatase domain. **b** Phylogenetic conservation of the depicted domain architecture among fungi. The *P*-score is a measure for peroxisomal targeting probability based on the C-terminal dodecamer[64]. **c** Fluorescence microscopic pictures of wild-type (WT) yeast cells expressing either Ptc5-RFP or Ptc5-RFP-PTS (magenta) together with the peroxisomal membrane protein Ant1-YFP (cyan). White color indicates colocalization. Scale bar represents 5 μm. **d** Intracellular localization of Ptc5-RFP-PTS and Ant1-YFP in Δ*pex5* cells. White color indicates colocalization. Scale bar represents 5 μm. **e** Mitochondrial (Mito) and peroxisomal (Px) fractions of strains expressing GFP-Sps19 and either Ptc5-RFP or Ptc5-RFP-PTS were prepared by density gradient centrifugation and analyzed by western blot (also see Supplementary Fig. 1e). GFP-Sps19 is a peroxisomal matrix protein and Por1 is located in the outer mitochondrial membrane. Source data are provided in the Source data file.

(ΔMTS Ptc5$^{1-201}$-RFP-PTS) localized to peroxisomes, but was not cleaved (Fig. 4c, d). This indicates that the IMP complex is active only in mitochondria but not in peroxisomes. Together these data show that Ptc5 is targeted to mitochondria and processed by the IMP machinery prior to its translocation into the peroxisomal matrix (Fig. 5a).

**A molecular tug-of-war-like mechanism.** In principle, different mechanisms for this indirect sorting to peroxisomes could be imagined. One possibility would be the release of processed Ptc5 from mitochondria back into the cytosol and subsequent import into peroxisomes. Retro-translocation into the cytosol has been observed for several mitochondrial proteins[24–26]. However, we did not observe cytosolic localization of Ptc5-RFP-PTS in Δ*pex5* mutants or in other mutants defective for peroxisomal import (Fig. 1d, Supplementary Fig. 1c). Also, Ptc5-RFP lacking the PTS1 was not detected in the cytosol (Fig. 1c). Expression of cytosolic Ptc5$^{Δ1-83}$-RFP mimicking the processed form of Ptc5-RFP

resulted in stable and clearly visible fluorescence in the cytosol (Fig. 5b). This also eliminates the possibility that the absence of cytosolic Ptc5 is due to its inherent instability in this compartment.

Alternatively, binding of Pex5 to Ptc5 during mitochondrial import could mediate a direct transfer from mitochondria to peroxisomes (Fig. 5a). This mechanism is supported by the observation that replacing the PTS1 of Ptc5 with PTS1 motifs of lower import efficiency[11] shifted the distribution of Ptc5 toward mitochondria (Supplementary Fig. 4f). Thus, Pex5 appears to bind the C-terminus of Ptc5 during mitochondrial import. Subsequent interaction of this complex with the peroxisomal import machinery likely pulls the processed protein out of the mitochondrial intermembrane space to facilitate the translocation into peroxisomes (Fig. 5a)

**Ptc5$^{ΔTM}$ increases association of peroxisomes and mitochondria.** Next, we asked if import into peroxisomes also occurs if

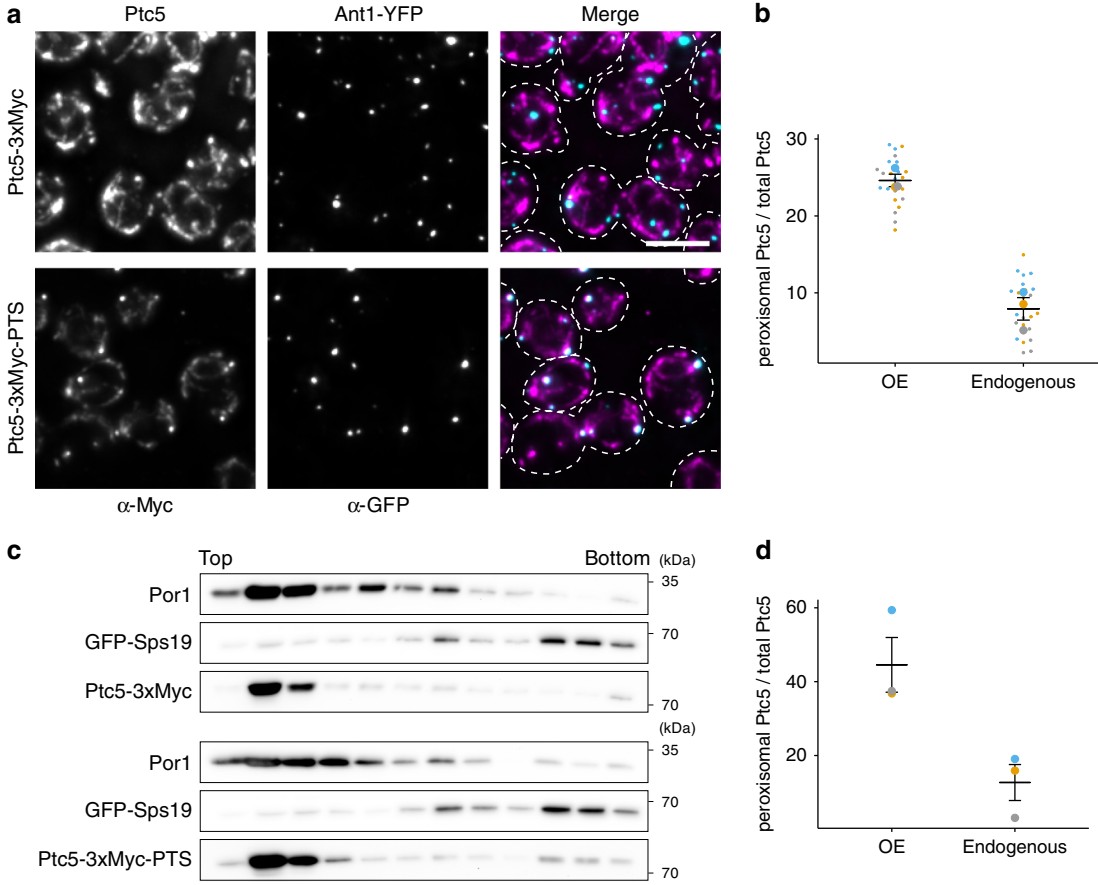

**Fig. 2 Dual targeting of endogenously tagged Ptc5. a** Yeast cells expressing either C-terminally (upper panel) or internally (lower panel) 3xMyc tagged Ptc5 (magenta) together with Ant1-YFP (cyan) were analyzed by immunofluorescence microscopy. Antibodies against Myc and GFP were used for detection. White color indicates colocalization in merged pictures. Scale bar represents 5 μm. **b** The fraction of peroxisomal Ptc5 signal to total Ptc5 signal was quantified using ImageJ[59]. 'OE' indicates Ptc5-RFP-PTS under control of the *tef1* promoter (Fig. 1c). 'Endogenous' refers to Ptc5-3xMyc-PTS (**a**). Data of three independent experiments (*n* = 3; each color represents one experiment) were plotted. Error bars represent standard error of the mean. **c** Organelles prepared from strains co-expressing either Ptc5-3xMyc (upper three western blots) or Ptc5-3xMyc-PTS (lower three western blots) together with the peroxisomal matrix protein GFP-Sps19 were analyzed by density gradient centrifugation and western blot. Por1 is a protein localized in the outer mitochondrial membrane. Twelve fractions were collected, starting from the top of the gradient, and analyzed. **d** Western blots from three independent experiments (*n* = 3; each color represents one experiment) were quantified using ImageJ to determine the ratio of peroxisomal to total Ptc5 signal of Ptc5-RFP-PTS (OE) and Ptc5-3xMyc-PTS (endogenous), respectively. Error bars represent standard error of the mean. Source data are provided in the Source data file.

Ptc5 is initially targeted to the mitochondrial matrix. Therefore, we investigated sorting of a Ptc5 variant lacking the transmembrane domain, but retaining the mitochondrial presequence (Ptc5$^{\Delta TM}$-RFP-PTS). This fusion protein predominantly localized in mitochondria and accumulated in distinct mitochondrial foci frequently decorated with a peroxisome (Fig. 5c, d). Hence, the association of peroxisomes and mitochondria in strains expressing Ptc5$^{\Delta TM}$-RFP-PTS or Ptc5$^{\Delta TM}$-RFP together with Tim50-CFP and Ant1-YFP was analyzed (Fig. 5c, d, Supplementary Fig. 5a). Expression of Ptc5$^{\Delta TM}$-RFP-PTS significantly increased the number of peroxisomes associated with mitochondria (Fig. 5e). Removal of the PTS1 or deletion of *pex5* resulted in mitochondrial localization and reduced association of the organelles (Fig. 5c–e). Thus, two import complexes pulling the protein in opposite directions control the intracellular distribution of Ptc5 (Fig. 5a, e). To test a similar type of interaction between a different pair of organelles we fused a PTS1 to Sec63-RFP (ref. [27]), which is localized in the membrane of the endoplasmic reticulum (ER). Expression of Sec63-RFP-PTS resulted in decoration of stained ER foci with peroxisomes (Supplementary Fig. 5b, c).

These data strengthen the idea that competing targeting signals can increase the proximity of different organelles.

**Gpd1 is a target of peroxisomal Ptc5.** Phosphoproteomics experiments have identified glycerol-3-phosphate dehydrogenase Gpd1 as a substrate of Ptc5[16]. Gpd1 participates in peroxisomal NAD/NADH homeostasis, and resides both in the cytosol and in peroxisomes[28,29]. We enriched peroxisomes via density gradient centrifugation (Supplementary Fig. 6a). We used Phos-tag[30] SDS–PAGE to analyze the phosphorylation status of peroxisomal Gpd1-GFP. Deletion of *ptc5* led to the detection of phosphorylated Gpd1-GFP migrating with reduced mobility (Fig. 5f). λ-phosphatase treatment of phosphorylated Gpd1-GFP from Δptc5 samples resulted in an electrophoretic mobility identical to Gpd1-GFP from WT samples. As an additional control, we separated proteins from WT and Δptc5 cells on a standard SDS–PAGE and detected no differences in migration (Fig. 5g). This reveals that Ptc5 dephosphorylates Gpd1 inside peroxisomes. Dephosphorylation increases the enzymatic activity of Gpd1 (ref. [31]), suggesting that peroxisomal import of Ptc5 results in activation of Gpd1.

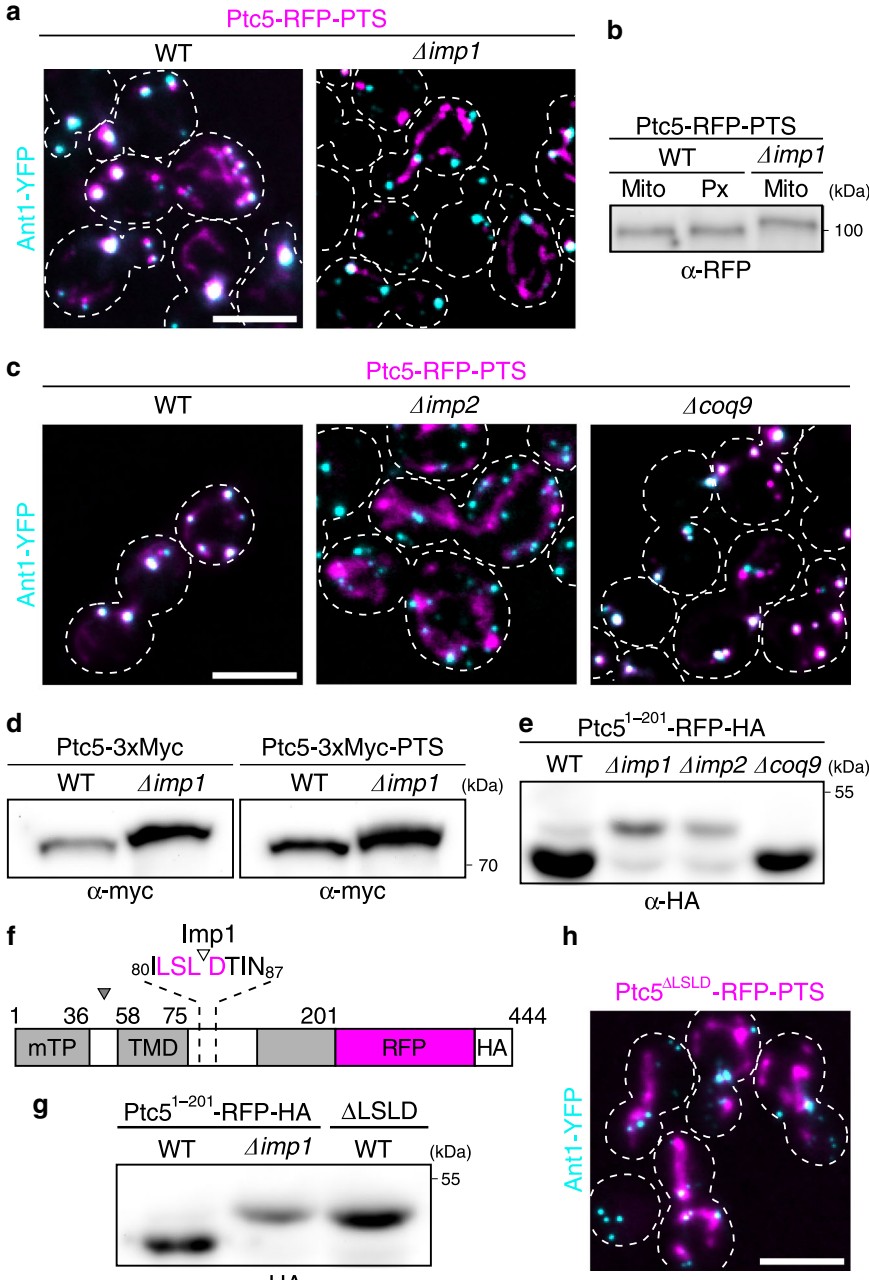

**Fig. 3 Ptc5 transits mitochondria en route to peroxisomes. a** Intracellular localization of Ptc5-RFP-PTS (magenta) and Ant1-YFP (cyan) in WT and *Δimp1* cells. White color indicates colocalization. Scale bar represents 5 μm. **b** Ptc5-RFP-PTS in mitochondrial (Mito) and peroxisomal (Px) fractions derived from gradient centrifugation (Supplementary Fig. 2a) was detected by western blot. A mitochondrial fraction (Supplementary Fig. 2a) from *Δimp1* cells was analyzed to detect unprocessed Ptc5-RFP-PTS. **c** Localization of Ptc5-RFP-PTS (magenta) was investigated by fluorescence microscopy in indicated strains. Ant1-YFP (cyan) served as marker for peroxisomes. White color indicates colocalization. Scale bar represents 5 μm. **d** Migration of Ptc5-3xMyc and Ptc5-3xMyc-PTS isolated from WT, and *Δimp1* cells was investigated using SDS–PAGE and western blot. **e** The truncated Ptc5 variant Ptc5$^{1-201}$-RFP-HA from indicated strainswas analyzed by SDS–PAGE and western blot. Truncation increased resolution between higher and lower mobility bands. **f** Schematic representation of the Imp1 cleavage site in Ptc5$^{1-201}$-RFP-HA as identified by mass spectrometry (see Supplementary Table. 1). The cleavage site is located between amino acid residue 83 and 84. **g** Processing of Ptc5$^{1-201}$-RFP-HA was analyzed by SDS–PAGE and western blot. ΔLSLD refers to a construct derived from Ptc5$^{1-201}$-RFP-HA lacking the amino acids surrounding the cleavage site (see Fig. 2f). **h** Intracellular localization of Ptc5$^{ΔLSLD}$-RFP-PTS and Ant1-YFP. White color indicates colocalization. Scale bar represents 5 μm. Source data are provided in the Source data file.

**Indirect targeting of Ptc5 prevents premature activity**. We noticed that mis-targeting of Ptc5 to the cytosol (Ptc5$^{Δ1-83}$-RFP) resulted in poor cell growth (Fig. 5h). Expression of cytosolic phosphatase-dead variants (D302A and D424A, respectively) did not affect growth (Fig. 5i), although expression levels were higher compared to the progenitor protein Ptc5$^{Δ1-83}$-RFP and

localization was unchanged (Fig. 5j, Supplementary Fig. 6b). Thus, direct transfer from mitochondria to peroxisomes may serve to avoid deleterious Ptc5 phosphatase activity in the cytosol. Peroxisomal targeting of Ptc5 (Ptc5$^{Δ1-83}$-RFP-PTS) without mitochondrial transit also caused a growth phenotype (Fig. 5h). This indicates that even transient residence of active Ptc5 in the

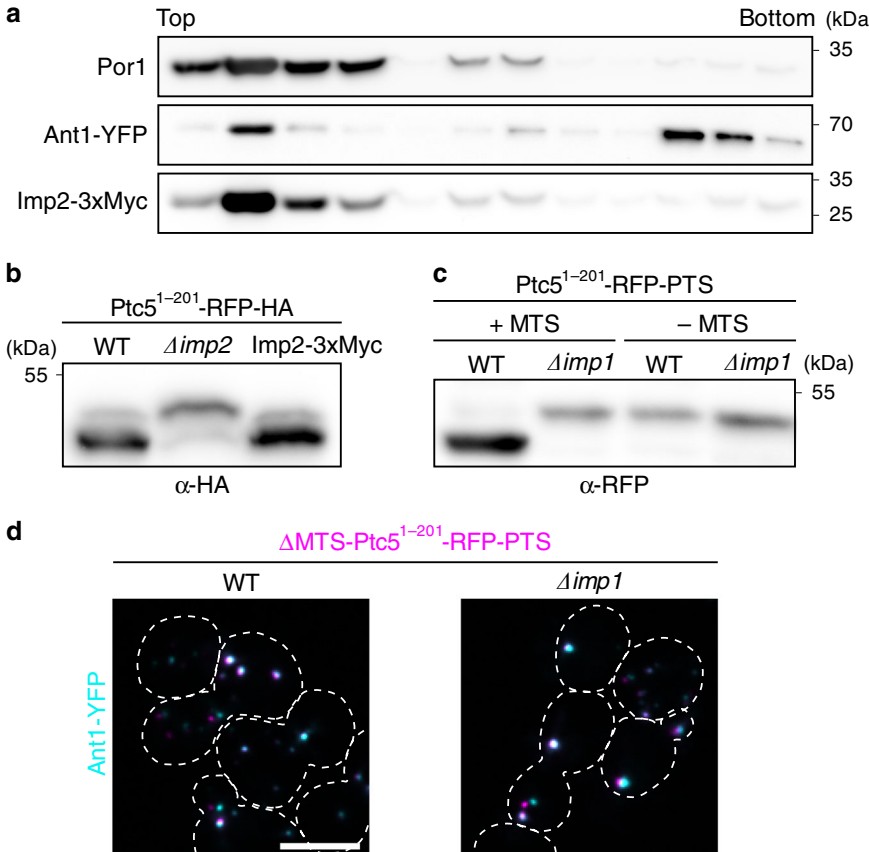

**Fig. 4 The IMP complex localizes to mitochondria but not to peroxisomes. a** Localization of Imp2-3xMyc was analyzed by density gradient centrifugation using purified organelles also containing the peroxisomal membrane protein Ant1 fused to YFP. Fraction were collected from the top of the gradient and analyzed by SDS–PAGE and western blot. Por1 is a protein located in the outer membrane of mitochondria. **b** Activity of Imp2-3xMyc was studied by analyzing the processing of the truncated Ptc5 variant Ptc5$^{1-201}$-RFP-HA. **c** Migration of Ptc5$^{1-201}$-RFP-PTS either containing (+MTS) or lacking (−MTS) the mitochondrial presequence was examined by SDS–PAGE and western blot. **d** The subcellular localization of ΔMTS Ptc5$^{1-201}$-RFP-PTS was investigated using fluorescence microscopy. White color indicates colocalization with the peroxisomal protein Ant1-YFP. Source data are provided in the Source data file.

cytosol is deleterious for the cells. The detour via mitochondria could prevent premature folding, since import into this organelle occurs only in the unfolded state and also co-translationally[1,32].

**Additional mitochondrial proteins with a PTS1.** Our experiments uncovered a remarkable interaction between import machineries of two different cellular compartments. We identified putative PTS1 motifs in several previously validated mitochondrial proteins[33] (Supplementary Table 1). These candidate proteins lack a transmembrane domain and are no substrates of the IMP complex. However, their domain structure may allow for simultaneous interaction with import complexes of mitochondria and peroxisomes. Three out of four tested candidate proteins exhibited a dual localization (Supplementary Figs. 7 and 8). Pxp2 is a protein of unknown molecular function, whose peroxisomal localization has been previously described[14]. Of interest, deletion of *pxp2* leads to growth defects on non-fermentable carbon sources, such as oleate and ethanol[14]. Pxp2-RFP-PTS accumulated in foci at the interface of peroxisomes and mitochondria, but was also found in peroxisomes not in proximity to mitochondria (Fig. 6a, b). Pxp2-RFP was localized in foci only overlapping with Tim50-YFP but not with Ant1-YFP (Fig. 6a). Density gradient centrifugation experiments confirmed dual localization of Pxp2-RFP-PTS and indicate that Pxp2-RFP foci are located inside mitochondria (Fig. 6c). The N-terminus of Pxp2 was found to contain a functional mitochondrial

presequence (Fig. 6d). Similar to Ptc5$^{\Delta TM}$-RFP-PTS, Pxp2-RFP-PTS also increased the association of peroxisomes and mitochondria (Fig. 6e). Expression of the mitochondrial inner membrane protein Mss2-RFP-PTS (ref. [34]) and the protein of unknown function Dpi8-RFP-PTS had a positive effect on the number of peroxisome in proximity to mitochondria (Supplementary Fig. 7d, e). Dpi8-RFP-PTS was regularly distributed inside mitochondria but also localized in peroxisomes. Remarkably, several cells only contained peroxisomal signal of Dpi8-RFP-PTS, while other cells contained predominantly mitochondrial signal (Supplementary Figs. 7 and 8). Thus, several *S. cerevisiae* proteins carrying competing targeting signals could induce an interaction between mitochondria and peroxisomes by a tug-of-war-like mechanism.

**Discussion**
In this study, we have identified an indirect pathway for protein targeting to peroxisomes via mitochondria. While previously characterized mechanisms for dual targeting to peroxisomes and mitochondria mostly involve the formation of distinct protein isoforms[11], sorting of Ptc5 is different. It requires mitochondrial processing at the inner membrane and coincident interaction with the import complexes of both compartments. This mechanism allows cells to compensate for a specific limitation of peroxisomal matrix proteins containing a PTS1, which can only be recognized by Pex5 after translation. This enables folding and

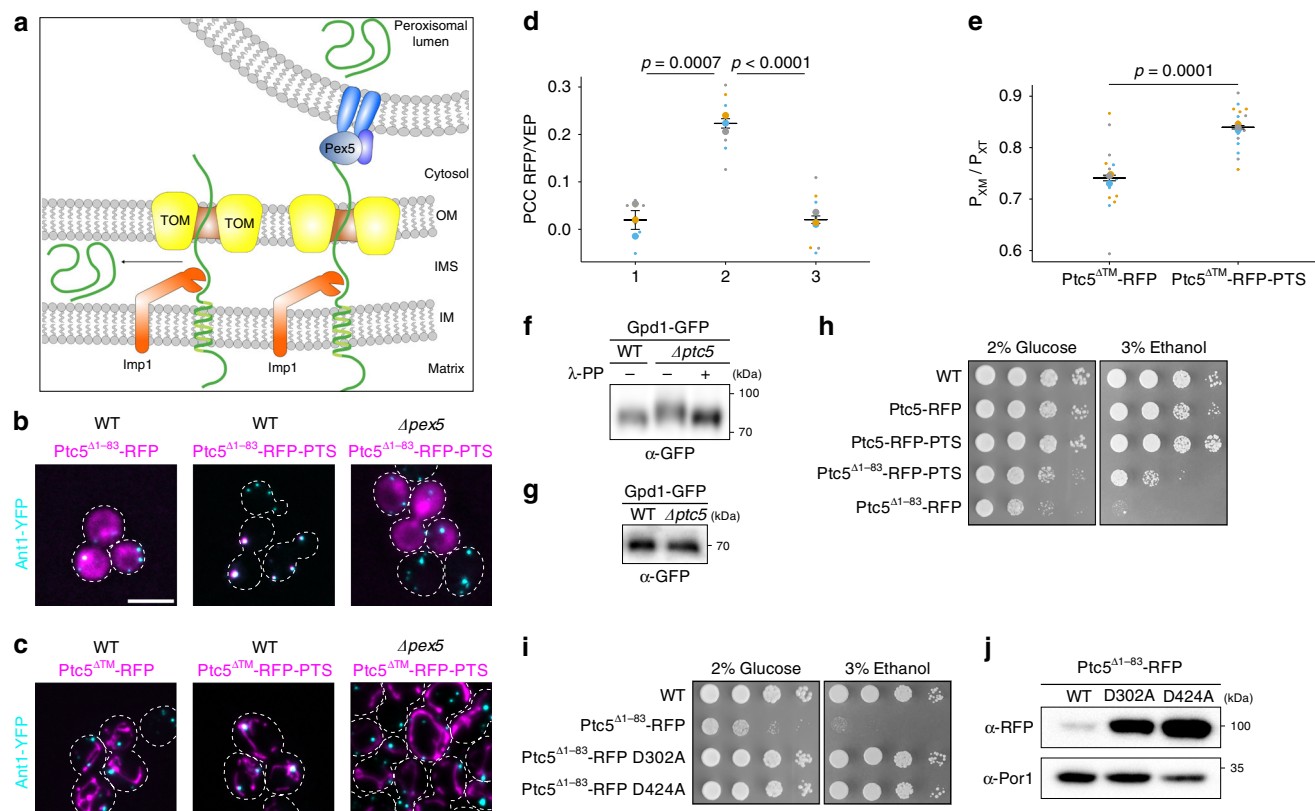

**Fig. 5 A tug-of-war-like mechanism sorts Ptc5 from mitochondria to peroxisomes and prevents detrimental phosphatase activity in the cytosol.**
**a** Working model of the mechanism of Ptc5 sorting. Ptc5 (green) either localizes to the mitochondrial intermembrane space or translocates into the peroxisomal matrix after processing by Imp1 (orange). Translocation depends on Pex5 and other components of the peroxisomal protein import machinery (blue). **b** Fluorescence microscopic pictures of yeast cells expressing either Ptc5$^{\Delta 1-83}$-RFP or Ptc5$^{\Delta 1-83}$-RFP-PTS (magenta) together with the peroxisomal membrane protein Ant1-YFP (cyan) in indicated strains. White color indicates colocalization. Scale bar represents 5 μm. **c** Ptc5$^{\Delta TM}$-RFP or Ptc5$^{\Delta TM}$-RFP-PTS were co-expressed with Ant1-YFP in indicated strains. Subcellular localization was determined by fluorescence microscopy. White color indicates colocalization. Scale bar represents 5 μm. **d** Correlation between the signal of Ptc5$^{\Delta TM}$-RFP or Ptc5$^{\Delta TM}$-RFP-PTS and of Ant1-YFP was quantified. PCC refers to the Pearson's correlation coefficient. 1: Ptc5$^{\Delta TM}$-RFP in WT, 2: Ptc5$^{\Delta TM}$-RFP-PTS in WT, and 3: Ptc5$^{\Delta TM}$-RFP-PTS in Δpex5. Quantifications are based on $n = 3$ independent experiments. Each color represents one experiment. Error bars represent standard error of the mean. $P$-values were calculated with a two-tailed unpaired Student's $t$-test. **e** Quantification of the fraction of peroxisomes contacting mitochondria (Px$_M$) in relation to the total peroxisome count (Px$_T$) of cells co-expressing Ptc5$^{\Delta TM}$-RFP or Ptc5$^{\Delta TM}$-RFP-PTS with Ant1-YFP, and the mitochondrial membrane protein Tim50 fused to CFP (Supplementary Fig. 5a). Association of Ant1-YFP containing foci with Tim50-CFP was counted in three independent experiments ($n = 3$). Independent experiments are represented by different colors. Error bars represent standard error of the mean. $P$-values were calculated with a two-tailed unpaired Student's $t$-test. **f** PhosTag SDS–PAGE of peroxisomal fractions derived from WT and Δptc5 cells expressing Gpd1-GFP (Supplementary Fig. 6a) was analyzed by western blot to determine the phosphorylation status of Gpd1-GFP. λ-PP indicates treatment with λ-protein phosphatase prior to PhosTag SDS–PAGE. **g** Samples shown in **f** were also analyzed by conventional SDS–PAGE and western blot to assure that migration differences result from altered phosphorylation. **h, i** Serial dilutions of indicated strains were spotted on SC-HIS medium containing either glucose or ethanol as carbon source. **j** Western blot showing the expression levels of Ptc5$^{\Delta 1-83}$-RFP and its phosphatase-dead derivatives D302A and D424A. Por1 served as a loading control. Source data are provided in the Source data file.

biological activity preceding import into peroxisomes. Ptc5, which is toxic in the cytosol, can reach the peroxisomal lumen to dephosphorylate Gpd1 without folding in the cytosol via a detour through mitochondria.

Characterization of Ptc5 sorting added an additional layer of complexity to the molecular interplay of peroxisomes and mitochondria. A similar transport pathway might also operate in mammalian cells, as a recently characterized isoform of acyl-CoA-binding domain containing enoyl-CoA-Δ-isomerase is dually targeted to peroxisomes and mitochondria by competing targeting signals[35]. The metabolism of both organelles is tightly connected and they exchange various metabolites[36,37]. In addition, they share components of their fission machinery in yeast and mammalian cells, and use the same AAA-ATPase for

clearance of tail-anchored membrane proteins in yeast[38–44]. Recently, it was shown that mitochondria derived vesicles contribute to de novo peroxisome biogenesis in mammalian cells putting even more emphasis on the close relationship of both compartments[45].

Understanding the molecular mechanisms that enable physical interaction of organelles has emerged as an important research topic in the past decade[46,47]. While proteins mediating contact of peroxisomes with several organelles have been functionally characterized, knowledge about tethers between peroxisomes and mitochondria is still limited[48–52]. Systematic screens for peroxisome–mitochondria contact sites in S. cerevisiae identified two proteins with tethering functions, but also indicated that contact site forming proteins are functionally redundant[53]. We

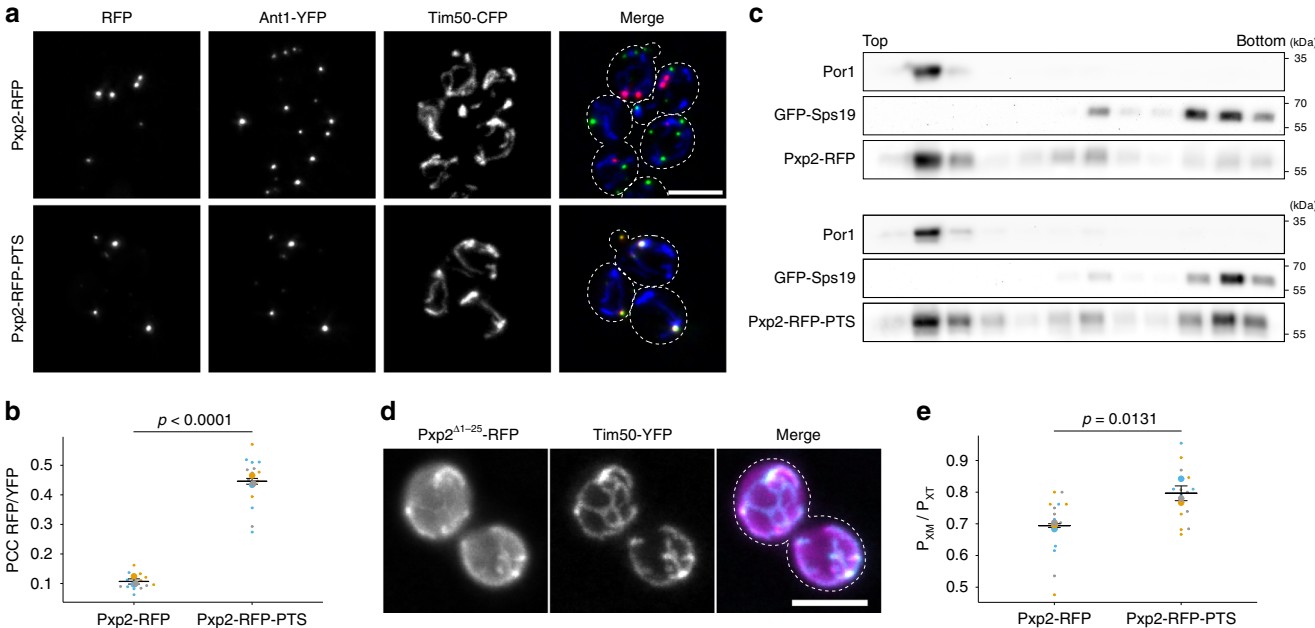

**Fig. 6 Dual targeting of Pxp2 to mitochondria and peroxisomes. a** Subcellular localization of Pxp2-RFP (red) and Pxp2-RFP-PTS (red) was determined using fluorescence microscopy. Ant1-YFP (green) and Tim50-CFP (blue) were used to label peroxisomes and mitochondria, respectively. White color represents overlap of all three signals. Scale bar represents 5 µm. **b** Correlation of Pxp2-RFP or Pxp2-RFP-PTS with Ant1-YFP was analyzed. PCC refers to Pearson's correlation coefficient. Quantifications are based three independent experiments ($n = 3$). Each experiment is represented by a different color. Error bars represent standard error of the mean. P-values were calculated with a two-tailed unpaired Student's t-test. **c** Subcellular localization of Pxp2-RFP and Pxp2-RFP-PTS was determined using density gradient centrifugation. Twelve fractions, collected from the top of the gradient, were analyzed by SDS–PAGE and western blot. **d** Twenty five amino acids from the N-terminus of Pxp2 were fused to RFP. Pxp2$^{1-25}$-RFP (magenta) was expressed in a strain also containing Tim50-YFP (cyan). The resulting strain was analyzed by fluorescence microscopy. White color indicates colocalization. Scale bar represents 5 µm. **e** Quantification of the fraction of peroxisomes contacting mitochondria (Px$_M$) in relation to the total peroxisome count (Px$_T$) of cells co-expressing Pxp2-RFP or Pxp2-RFP-PTS with Ant1-YFP, and the mitochondrial membrane protein Tim50 fused to CFP. Association of Ant1-YFP containing foci with Tim50-CFP was counted. Quantifications are based on three independent experiments ($n = 3$). Each experiment is represented by a different color. Error bars represent standard error of the mean. P-values were calculated with a two-tailed unpaired Student's t-test. Source data are provided in the Source data file.

observed an increase in the number of peroxisomes associated with mitochondria upon overexpression of the artificial Ptc5$^{\Delta TM}$-RFP-PTS or the other proteins with competing targeting signals. Thus, proteins mediating a tug-of-war-like interaction between mitochondria and peroxisomes may contribute to the redundancy of factors involved in organelle contact site formation.

## Methods

**Yeast strains, plasmids, and oligonucleotides**. Yeast strains used in this study are listed in Supplementary Table 2, and were derived from BY4741 or the Euroscarf gene deletion strains. Yeast deletions and insertions were generated by targeted PCR disruption using chemical transformation[54,55]. The following plasmids were used for PCR amplification[20,56]: pYM1, pYM4, pYM-N9, pYM25, and pCR125. pYM-3xHA-PTS and pYM-3xMyc-PTS were generated by inserting the PCR fragment using primers MK365/MK366 or MK365/ML242 into the XmnI/AscI fragment of pYM1 or pYM4, respectively. Replicative plasmids and oligonucleotides are listed in Supplementary Table 2.

**Yeast media and growth conditions**. Cells were grown in either YPD medium (1% yeast extract, 2% peptone, and 2% glucose) or synthetic complete minimal medium at 30 °C. For microscopic analysis and protein preparation cells from logarithmic growth phase were used. For peroxisome induction, cells were grown in YP medium containing 0.1% glucose overnight, washed once with sterile water, and resuspended in YNBO medium[57]. For growth assays, overnight cultures from SC medium lacking the respective supplements were washed with sterile water, adjusted to identical optical density, and 5 µl of serial dilutions were spotted onto SC medium containing 2% (w/v) glucose or 3% (v/v) ethanol. Plates were incubated at 30 °C for 2–4 days.

**Preparation of post-nuclear supernatants**. Post-nuclear supernatants (PNS) were prepared using a modified protocol from Cramer et al. (steps 1–16)[58]. Cells were precultured in 500 ml YP medium supplemented with 0.1% glucose at 30 °C

overnight. Cells were harvested at $6000 \times g$ for 6 min, washed once with 30 ml sterile water, and resuspended in 500 ml YNBO medium. After incubation at 30 °C for 16 h, cells were harvested as described above, washed twice with 30 ml sterile water, and incubated in 15 ml DTT buffer (100 mM Tris, 10 mM DTT, pH 7.4) for 30 min at 30 °C with gentle agitation. Cells were pelleted by centrifugation at $600 \times g$, washed three times with 15 ml sorbitol buffer (20 mM HEPES, 1.2 M sorbitol), and incubated with 15 ml sorbitol buffer containing 20 mg Zymolyase 100 T (Roth) at 30 °C with gentle agitation. Digestion of yeast cell walls was monitored photometrically at $OD_{600}$ and digestion was stopped when at least half of the cells lysed upon addition of water. From this point all steps were carried out on ice or at 4 °C. Spheroplasts were gently washed three times with 15 ml sorbitol buffer, resuspended in 15 ml lysis buffer (5 mM MES, 0.5 mM EDTA, 1 mM KCl, 0.6 M Sorbitol, 1 mM 4-aminobenzamidine-dihydrochloride, 1 µg/ml aprotinin, 1 µg/ml leupeptin, 1 mM phenylmethylsulfonyl fluoride, 10 µg/ml N-tosyl-L-phenylalanine chloromethyl ketone, and 1 µg/ml pepstatin), and frozen at −80 °C overnight. Spheroplasts were thawed on ice and homogenized using a Potter-Elvehjem homogenizer ($2 \times 12$ strokes). Nuclei and cell debris were removed by two subsequent centrifugations at $1600 \times g$ for 10 min. Subsequently, the PNS was diluted to an $OD_{600}$ of 1, aliquoted and frozen at −80 °C.

**Subcellular fractionation**. A total of 200 µl of PNS or PNS without cytosol were loaded onto a NycoDenz density gradient consisting of 333 µl 20%, 666 µl 25%, 666 µl 30%, and 333 µl 35% NycoDenz in gradient buffer A (5 mM MES, 1 mM EDTA, 1 mM KCl, and 0.1% (v/v) ethanol). Gradients were centrifuged in a Beckman L7-65 ultracentrifuge equipped with a Sorwall TST 60.4 rotor at $100,000 \times g$ (35,000 rpm) for 90 min at 4 °C. A total of 183 µl fractions were collected from the top of the gradient and either directly used for SDS–PAGE or stored at −80 °C. In case of low protein concentration, samples were subjected to TCA precipitation prior to analysis. Briefly, TCA was added to a final concentration of 20%. Samples were kept at 4 °C overnight, washed two times with ice-cold acetone, and resuspended in 2x SDS sample buffer. For the Gpd1-GFP experiment, PNS fractions were subjected to an additional $13,000 \times g$ spin (5 min) and pellets were dissolved in lysis buffer buffer (see above). Fractions subjected to PhosTag SDS–PAGE were prepared with gradient buffer A lacking EDTA (ref. [57]).

For gradients based on glucose grown cells (Supplementary Fig. 2a), the PNS was washed with urea as follows: the PNS was pelleted at $13,000 \times g$ for 10 min at 4 °C, washed once with lysis containing 1.5 M urea, centrifuged at $13,000 \times g$ for 10 min at 4 °C, and resuspended in lysis buffer.

**Epifluorescence microscopy.** A total of 200 μl of hot 1.5% agarose melted in water was used to create a thin agarose cushion on a 76 × 26 mm microscope slide (Roth, Karlsruhe, Germany). Cells were washed with water, concentrated tenfold, 3 μl were spotted onto the middle of the agarose pad and covered with an 18 × 18 mm coverslip (Roth, Karlsruhe, Germany). Microscopy was performed on an Axiovert 200 M inverse microscope (Zeiss) equipped with a 1394 ORCA ERA CCD camera (Hamamatsu Photonics), filter sets for cyanGFP, enhanced GFP, (EGFP), yellow fluorescent protein (YFP), and rhodamine (Chroma Technology, Bellows Falls, VT), and a Zeiss 63× Plan Apochromat oil lens (NA 1.4). Single-plane bright field or phase contrast images and z-stacks of the cells (0.5 μm z-spacing) in the appropriate fluorescence channels were recorded, using the image acquisition software Volocity 5.3 (Perkin-Elmer). Images were processed and evaluated in ImageJ[59]. For protein localization analysis, z-projections of deconvolved image stacks of the fluorescent channels were used. Deconvolution was performed on the z-stacks by the ImageJ plugin DeconvolutionLab with 25 iterations of the Richardson–Lucy algorithm. Pearson's correlation coefficients were determined with Volocity 5.3.

**Immunoblotting and antibodies.** Proteins were extracted as described by Kushnirov (2000) using a modified sample buffer (50 mM Tris-HCl, pH 6.8, 2% SDS, 6% glycerol, 0.025% bromophenol blue, and 50 mM dithiothreitol)[59]. In brief, 1 OD$_{600}$ of yeast cells were pelleted at $13,000 \times g$ for 1 min and incubated with 300 mM 0.2 M NaOH for 5 min at room temperature. Cells were pelleted, resuspended in 50 μl sample buffer, incubated for 5 min at 95 °C, pelleted, and the supernatant was transferred to a new reaction tube. Proteins were incubated for 5 min at 95 °C at 750 rpm prior to loading. SDS–PAGE was performed with self-cast or Midi-protean TGX precast 7.5% gels (BioRad), PageRuler Prestained protein ladder (Thermo Fisher) as protein standard and a BioRad Mini- or Midi-Protean cell. Proteins were then blotted onto a PVDF membrane in a semi-dry blotter (Peqlab) at 75 mA per mini-gel for 2 h. Membranes were blocked for 30 min in TBST containing 5% nonfat dry milk (Roth) and incubated in primary antibody containing 0.05% sodium azide with gentle agitation for 2 h at room temperature or overnight at 4 °C. After removal of the antibody solution, membranes were washed three times with TBST for 5 min, incubated with HRP-conjugated secondary antibody for 45 min at room temperature, washed three times with TBST for 5 min, and then developed using either Pierce ECL Western Blotting Substrate (Thermo) or SuperSignal West Femto Maximum Sensitivity Substrate (Thermo). The following antibodies were used in this study: anti-GFP (1:5000, TP401, Torrey Pines Biolabs), anti-HA (1:2500, ab1302275, Abcam), anti-tagRFP (1:1000, AB233, Evrogen), anti-Por1 serum (1:1000, kindly provided by Roland Lill, Marburg), anti-Myc (1:1000, Cell Signaling Technology), m-IgGκ BP-HRP (1:5000, sc-2005, Santa Cruz Biotechnology), and mouse anti-rabbit IgG-HRP (1:5000, sc-2357, Santa Cruz Biotechnology).

**Carbonate extraction.** A protocol published by Fujiki et al. was followed with slight modifications[60]. A total of 300 μl of PNS were centrifuged at $13,000 \times g$ for 10 min at 4 °C, pellets were resuspended in 300 μl lysis buffer (control samples) or 300 μl lysis buffer containing 100 mM sodium carbonate (freshly prepared) and incubated for 15 min on ice. After centrifugation at $110,000 \times g$ for 60 min at 4 °C, the supernatant was separated from the pellet, the pellet was rinsed once with lysis buffer, and dissolved in Thorner buffer (40 mM Tris-HCl, pH 8, 5% SDS, 8 M urea, 100 μM EDTA, and 50 mM dithiothreitol). Samples were analyzed by SDS–PAGE and western blot.

**Immunoprecipitation.** Cells (10,000 OD$_{600}$ per strain) were pelleted at $6000 \times g$ for 10 min at room temperature, resuspended in 10 ml lysis buffer (20 mM Tris-HCl pH 8, 137 mM NaCl, 1% Triton X-100, 2 mM EDTA, 1 mM 4-aminobenzamidinedihydrochloride, 1 μg/ml aprotinin, 1 μg/ml leupeptin, 1 mM phenylmethylsulfonyl fluoride, 10 μg/ml N-tosyl-L-phenylalanine chloromethyl ketone, and 1 μg/ml pepstatin), and ground to a fine powder using a pestle and mortar filled with liquid nitrogen. After thawing, extracts were centrifuged at $1300 \times g$ for 10 min at 4 °C and subsequently incubated with 25 μl Pierce Anti-HA magnetic beads (Thermo Fisher) according to the manufacturer's instructions. After washing, beads were resuspended in 20 μl protein loading buffer and incubated at 95 °C for 10 min to elute bound protein. Beads were removed by centrifugation prior to sample loading.

**Silver staining.** The procedure was based on a protocol from Mortz et al.[61]. After electrophoresis, gels were fixed with 40% ethanol, 10% acetic acid, 50% H$_2$O for 1 h, and then washed once with water for 30 min. Gels were sensitized using 0.02% sodium thiosulfate for 1 min, washed three times with water for 1 min, and incubated for 20 min at 4 °C with prechilled 0.1% silver nitrate solution containing 0.02% formaldehyde. Gels were washed three times with water for 1 min, placed in a new staining tray, washed again with water for 1 min, and then developed using

3% sodium carbonate solution containing 0.05% formaldehyde. Staining was terminated using 5% acetic acid and gels were stored in 1% acetic acid at 4 °C.

**Mass spectrometry.** Excised gel pieces were chopped into small pieces, destained with 30% isopropyl alcohol containing 60 mM ammonium carbonate and 30 mM thioglycolic acid, dehydrated with isopropyl alcohol, and dried. Gel pieces were rehydrated in 10% acetonitrile containing 5 mM ammonium bicarbonate, 8 mM DTT and 2.5 μg/ml sequencing grade modified trypsin (Promega), and incubated overnight at 30 °C. Samples were desalted using C18 microspin columns (Nest Group) according to the manufacturer's instructions. Dried and reconstituted peptides were analyzed by LC–MS using a Q-Exactive Plus instrument connected to an Ultimate 3000 RSLC nano and a nanospray flex ion source (Thermo Scientific). Peptide separation was performed on a reverse phase HPLC column (75 μm × 40 cm) packed with C18 resin (2.4 μm) using a formic acid/acetonitrile gradient. The data acquisition mode was set to obtain one high-resolution MS scan at a resolution of 70,000 full width at half maximum (at $m/z$ 200) followed by MS/MS scans of the most intense ions. MS/MS data were searched against a database containing the Ptc5$^{1-201}$-RFP-HA protein sequence using Mascot embedded into Proteom Discoverer 1.4 software (Thermo Scientific).

**Immunofluorescence staining.** Cells were grown to mid-log phase in YPD and 5 ml of culture were fixed by addition of 600 μl formaldehyde (4% f.c) for 60 min at 30 °C with agitation. After fixation, cells were washed twice with 0.1 M potassium phosphate buffer pH 6.5 (Phos) and twice with Phos containing 1.2 M sorbitol (Phos/Sorb). A total of 500 μl of fixed cells were supplemented with 10 μl 1 M beta-mercaptoethanol and 5 μl Zymolyase 100 T (100 mg/ml, Roth). Cells were incubated on a rotating wheel at 23 °C for 30 min. Meanwhile, glass bottom petri dishes (35 mm, MatTek) were coated with poly-L-lysine (0.1%, Sigma) for 30 min at RT, washed three times with deionized water, and air dried. Spheroplasts were harvested by centrifugation at $300 \times g$ for 3 min at 4 °C and gently washed two times with Phos/Sorb. Cells were resuspended in 200 μl Phos/Sorb, transferred to the coated petri dishes, and incubated for 30 min at RT. Phos/Sorb was removed and cells were permeabilized with 200 μl Phos/Sorb containing 1% Triton X-100 for 2 min at RT. Cells were washed twice with Phos/Sorb, four times with 1% BSA/PBS, and incubated with 200 μl primary antibody in BSA/PBS for 1 h at RT (anti-GFP, 1:500, TP401, Torrey Pines Biolabs; anti-HA, 1:250, ab1302275, Abcam; anti-Myc, 1:250, #2276, Cell Signaling Technology). Cells were washed five times with BSA/PBS and incubated with 200 μl secondary antibody in BSA/PBS for 45 min at RT in the dark (Alexa Fluor® 488 AffiniPure Donkey anti-rabbit, Alexa Fluor® 594 AffiniPure Donkey anti-mouse, both 1:200, Jackson ImmunoResearch). Cells were washed five times with PBS/BSA and imaged.

**Analysis of protein phosphorylation by PhosTag SDS–PAGE.** For PhosTag analysis of peroxisomal Gpd1-GFP, self-cast mini SDS gels (10 % acrylamide) containing 100 μM PhosTag and 100 μM MnCl$_2$ were used, and proteins were blotted onto PVDF membranes using a BioRad wet blotting system at 25 V overnight. For protein dephosphorylation, 20 μl of a peroxisomal fraction from a ptc5 mutant expressing Gpd1-GFP (see Supplementary Fig. 6a) were incubated with 0.5 μl λ-protein phosphatase for 60 min at 30 °C according to the manufacturer's protocol. Untreated samples were handled identically but contained 0.5 μl of heat-inactivated λ-protein phosphatase (65 °C for 60 min).

**Yeast two-hybrid analysis.** The sequence encoding the TPR domains of Pex5 was inserted into pGBKT7 (Matchmaker GAL4 Two-Hybrid System 3; Clontech). The ORFs for GFP, GFP-SKL, or GFP-PTS1$_{Ptc5}$ were cloned into pGADT7 (Matchmaker GAL4 Two-Hybrid System 3; Clontech). pGBKT7-Pex5$_{TPR}$ and derivatives of pGADT7 were co-transformed into YTS398. The transformed strains were grown in liquid SD medium, lacking leucine and tryptophan, to an OD$_{600}$ of 1.0. Serial dilutions of each strain were spotted on solid SD lacking leucine and tryptophan as growth control, and on SD medium lacking leucine, tryptophan, histidine to test for protein–protein interaction.

**Statistics and reproducibility.** Microscopic data was collected from three independent S. cerevisiae cultures. Five images per culture were quantified. Pearson's correlation coefficient was calculated with Volocity 5.5.2. Superplots[62] and Student's t-tests were computed using RStudio 1.2.1335 with R 3.6.0. Blots are structured as follows: center line, mean; error bars, standard error of the mean; big circles, mean of experiments; and small circles, data points of experiments. P-values were calculated using an unpaired, two-sided Student's t-test. For calculation of Pearson's correlation coefficients all analyzed images contained ten or more cells and one image represents one data point. For quantification of contacts between mitochondria and peroxisomes five cells per image per strain from three independent experiments were analyzed. Quantification was performed by manual inspection of cells showing a signal in the RFP channel. Inspection was carried out without knowledge of the respective genotypes. All experiments were at least repeated three times with similar results.

**Bioinformatics.** *S. cerevisiae* protein sequences were retrieved from the *Saccharomyces* genome database and manipulated with notepad++ using regular expressions (regex). Protein sequences containing a PTS1 were bookmarked with the regular expression '([SA][RK][LI]|[SA][RK][MFV])\*$' and unmarked lines were removed. TMHMM v 2.0 was used to analyze transmembrane domains[63]. High-confidence mitochondrial proteome data[33] were used to search for mitochondrial proteins with PTS1. Other protein sequences were retrieved from NCBI.

**Reporting summary.** Further information on research design is available in the Nature Research Reporting Summary linked to this article.

## Data availability

Any original data to support the findings in this study are available from the corresponding authors on request. Non-cropped western blots (Figs. 1e; 2c; 3b, d, e, g; 4a–c; 5f, g, j and 6c, and Supplementary Figs. 1e; 2a, b; 3; 4a; 4c–e and 6a) and data underlying all plots (Figs. 2b, d; 5d, e and 6b, e, and Supplementary Figs. 1b; 5c, d and 7b–e) are provided as Source data file. The mass spectrometry proteomics data have been deposited to the ProteomeXchange Consortium via the PRIDE partner repository with the dataset identifier PXD018591.

## Code availability

All the code generated for mining of data or statistical analysis is available from the authors on request.

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

## Acknowledgements
We thank Bob Lesch, Roland Lill, Chris Meisinger, Randy Schekman, and Christof Taxis for antibodies, plasmids, and yeast strains. We acknowledge Eden Yifrach and Einat Zalckvar for sharing their unpublished data. We are grateful to Julia Ast, Uwe Maier, and Kay Schink for critical reading of the manuscript. We thank Marisa Piscator for excellent technical assistance. This work was supported by the German Research Foundation (Collaborative Research Center 987 and grant nr. BO2094-5) and the LOEWE program of the state of Hesse. J.F. received a fellowship from the Leopoldina.

## Author contributions
J.F. and M.B. designed the study. T.S., J.F., and M.K. performed all experiments with the exception of LC/MS, which was performed by J.K. J.F., T.S., and M.B. analyzed the data, wrote, and revised the manuscript.

## Competing interests
The authors declare no competing interests.
