## [Peer Review File · Nature Communications]

Reviewers' comments:

Reviewer #1 (Remarks to the Author):

The report submitted by Stehlik et al. studies a mechanism for the targeting of a subset of proteins to peroxisomes following their translocation into the mitochondria. The main body of their work focuses on Ptc5, a protein phosphatase originally described as localising to the intermembrane space of the mitochondria, whereby they bioinformatically identify a putative type 1 peroxisomal targeting signal at its C-terminus. Through a set of imaging and biochemical experiments, using a range of truncation constructs, they propose the following model: Ptc5 is first imported into the mitochondria via its N-terminal targeting sequence, where it is cleaved by the IMP machinery between amino acids 83 and 84. Ptc5 can then associate with the peroxisomal import machinery (e.g. Pex5) to then translocate from the mitochondria to the peroxisomal lumen. In doing so, the deleterious effects of Ptc5 phosphatase activity residing in the cytoplasm can be avoided. The experiments supporting this model are well controlled and, overall, convincing. Their work adds further weight to the developing idea that there is a complex interplay between these two metabolic organelles.

The authors have provided a broad range of data in support of their model that Ptc5 can localise to both mitochondria and peroxisomes. However, whilst their data showing that their constructs can localise to both organelles are compelling, I'm a little worried that they don't definitively show that this would indeed occur with the wild-type protein.

Main points

1. Most of the evidence stems from the fact that their reporter constructs show evidence of cleavage by the IMP machinery in the mitochondria while accumulating in peroxisomes. Yet, it would be important to discount the possibility that some of the IMP peptidase resides in the peroxisome.
2. Much of the data relies on variants of the Ptc5-RFP-PTS construct. It is important to ensure that the RFP is not detrimental to full translocation into the mitochondria, allowing more time for the PTS1 to have a role in the localisation of the protein. The Ptc5-3xHA-PTS expressing yeast certainly go a long way in testing this possibility, as the protein is endogenously expressed and only has a small tag; however, from the representative images, this construct is far more localised to mitochondria than the RFP construct. This raises the worrisome possibility that much of the peroxisomal signal observed here is artifactual and due to the PTS lingering longer in the cytosol due to the time necessary to unfold and translocate the RFP. One way to test this is to quantify the amount of Ptc5-3xHA-PTS in the peroxisomal fraction and mitochondrial fractions
3. There are a couple of issues with the data studying a functional role for Ptc5 at peroxisomes. The band shift in Supp. Fig. 12b doesn't necessarily show hyperphosphorylation of Gpd1 in the Δ ptc5 cells. The addition of a λ -phosphatase treated sample and the inclusion of a "normal" SDS-PAGE blot (alongside the PhosTag gel) would control for many of the other potential causes of the band-shift observed. Furthermore, Supp. Fig. 12a is a little unclear. Currently it shows a massive enrichment of Gpd1 in the Por1-positive (i.e. mitochondrial) fractions, but the authors state it is localised to peroxisomes and cytoplasm. Both mitochondrial and peroxisomal markers might help to clear this up, as well as an indication of the lanes chosen for the PhosTag gel analysis.
4. Finally, an alternative model for the enzyme-dead mutants rescuing the growth defect observed in Fig. 3h-g is that these mutants are not expressed well, or that they no longer localise to the cytoplasm.

Minor points

1. It is concluded that Ptc5 Δ TM-RFP-PTS can induce contact between mitochondria and peroxisomes. Using confocal microscopy, it is not possible to conclude that the organelles are in fact contacting one another. As a result, the language would need to be softened.
2. Whilst the data on the dual localisation of Ptc5 are convincing, it isn't clear from the images provided that this is a general phenomenon for a subset of proteins (in this case: Pxp2, Mss2, Dpi8 and Rml2). Arguably, the images presented in Supp. Fig. 13&14 only show dual targeting for Dpi8, but the authors conclude that 3 out of 4 candidates showed this phenotype.
3. In the final remarks of the paper, the authors state that there is other work highlighting a close

relationship between peroxisomes and mitochondria, using Sugiura et al. (2017) to support their claim. The work shown in the referenced paper is based on a phenomenon that occurs in mammalian cells, but is not observed in yeast, i.e. PEX3 localising to mitochondria (indeed peroxisomal biogenesis is quite different between yeast and mammalian cells). Therefore, this is perhaps not the best evidence to support their point; certainly not on its own. Their claim, that there is a close relationship between the two compartments, arguably has substantial evidence in support of it, e.g. overlap in metabolic functions (oxidation of fatty acids, lipid synthesis, ROS metabolism); many proteins are shared – Fis1 (also shown in yeast), MUL1/MAPL, USP30, Miro1/2 (also shown in yeast, known as Gem1) etc.; they share the fission machinery. If true, the work shown in this manuscript is an important addition to this, as it shows that some proteins can transit through the mitochondria first. It might, therefore, be worth more comprehensively discussing this idea.

4. A schematic for the constructs generated, as well as stating the specific amino acid changes, would be useful.

Reviewer #2 (Remarks to the Author):

In this manuscript, Stehlik et al. report a new protein sorting pathway connecting mitochondria and peroxisomes. Specifically, the authors have identified several proteins that are targeted to the mitochondrial intermembrane space, processed by IMP (the mitochondrial inner membrane peptidase) and, finally, re-targeted to the peroxisome. Such a pathway is new and thus of considerable interest to all molecular/cellular biologists working on mitochondria and/or peroxisomes. With a few exceptions (see below), the data are solid. I have some questions and suggestions.

Major issue:

The data described in this work for Ptc5-RFP-PTS are rather interesting per se, and, as stated above, unveil a protein sorting pathway that was never described before. However, there may be some doubts regarding the biological relevance of these findings because the RFP moiety in the fusion protein may fold immediately after synthesis in the cytosol leading to an artificial arrest of the protein at the mitochondrial import sites (see for instance Rassow J et al. (1989) JCB 109:1421). It is highly unlikely that the same happens with the Ptc5-3HA-PTS protein. Therefore, the results obtained with this protein should be presented and discussed in the main body of the manuscript and not in supplementary fig 4.

Other issues:

Line 50 – reference to the experiment shown in Supplementary fig 1 – the mitochondrial localization of Ptc5-RFP-PTS (3 right panels) is hard to see. Please improve image quality

Line 62 – “...abolished sorting of Ptc5-RFP-PTS1 into peroxisomes”. There are a few “yellow” dots in Fig.1F, which I presume represent red and green labelling in different planes. If so, this should be acknowledged and explained.

Lines 62-63 – “Supplementary figure 5a”- I am not sure the Nycodenz gradients are really of help here. It is very difficult (if not impossible) to get a good separation between yeast mitochondria and peroxisomes. The same happens here. Thus, the experiment is not really conclusive (in the delta-imp1 strain, there is a small pool of Ptc5-RFP-PTS at the bottom of the gradient together with peroxisomes). Consider removing these data. Also, note that the triangles in the figure are not explained and “Ptc5-RFP” in the upper panel should be “Ptc5-RFP-PTS”

Line 70- “(supplementary Fig. 6)”- These data are very important and this experiment should be done correctly. At pH 11.5 (0,1 M Na₂CO₃) organelle membranes are disrupted yielding membrane fragments of low density (see Fujiki Y et al. (1982) JCB 93:97). Thus, they are

recovered by ultracentrifugation (100.000xg, 1 h.). A 13k spin for 10 min is really not enough and this is the reason why in the delta-imp1 strain significant amounts of both Ptc5-RFP-PTS and Por1 appear in the Na₂CO₃ S fraction.

Line 73 – the Deltacoq9 cells. Please provide a sentence to explain why these cells have a mitochondrial respiratory defect.

Line 82 – “supplementary table 1” – Maybe I missed it, but I could find no data in this table showing mapping of the IMP1 cleavage site on the Ptc5 protein.

Lines 105-109 – It seems that the authors are assuming that despite lacking a TM domain, this Ptc5 protein still localizes to the mitochondrial intermembrane space. Is this so? Shouldn't this protein end up in the mitochondrial matrix? Related to this issue: did the authors try protease-protection assays to show that mitochondrial Ptc5 proteins expose their C-termini into the cytosol?

Line 122- “hyperphosphorylated”. Why not just “phosphorylated”? Is a reference missing?

Lines 137-138 – The authors should provide a bit more information regarding the proteins shown in Fig. S13 and Fig. S14, and the results obtained (just as they did for Dpi8 in legend to Fig. S14).

Line 310 – “a, Membrane of indicted...”. The authors mean “a, Organelles of indicated...” , right? If not, I saw no reference to the preparation of membranes.

Line 458- A reference for the method is missing.

Supplementary fig 3 – It would be easier to understand this figure if the “top” and “bottom” of the gradients were indicated in the figure. Also, the small pool of Ptc5-RFP co-sedimenting with peroxisomes (GFP-Sps19) probably represents some mixed mitochondria/peroxisome aggregates (Por1 is also visible in these fractions). A small comment on this pool would also help to understand the results.

Reviewer #3 (Remarks to the Author):

In this manuscript, Stehlik and colleagues provide evidence that the type 2C protein phosphatase Ptc5 of yeast is dually localized between mitochondria and peroxisomes. Combining genetic, microscopic and biochemical methodologies, the authors provide strong initial, but not yet conclusive, evidence that Ptc5 is targeted first to the mitochondrial intermembrane space (IMS) by an N-terminal mitochondrial presequence that is cleaved in the IMS, and that some of the cleaved Ptc5 still in the mitochondrial translocon, where it remains linear in structure, is then pulled into the peroxisome via the activity of the peroxisomal targeting signal 1 (PTS1) receptor Pex5 that interacts with Ptc5's degenerate PTS1 (-PRL) at its carboxyl terminus. This all leads to import of Ptc5 into the peroxisomal matrix.

Although the initial evidence for the pathway of targeting of Ptc5 first to the mitochondrion and then to the peroxisome via the activity of Pex5 is strong, it is not conclusive. The authors have to:

- 1) provide time-course evidence, either by biochemical or microscopic pulse-chase analysis, that Ptc5 is first targeted to the mitochondrion and then directed to the peroxisomal matrix.
- 2) provide protein-protein interaction evidence (e.g. pull-downs) to show that Ptc5 directly interacts through its degenerate PTS1 with the PTS1 receptor Pex5.

Additional points the authors should address are:

Line 68. 'reduced mobility' NOT 'lower mobility'.

Line 82. 'These data were confirmed' NOT 'This data was confirmed'.

Lines 85-86. The data only 'suggest' but do not conclusively 'show' that "Ptc5 is first targeted to mitochondria, processed by the IMP machinery and then sorted to peroxisomes." (See Point 1 above.)

Lines 92-93. Cumbersome sentence structure. Rephrase as 'We failed to observe cytosolic localization of Ptc5-RFP-PTS either in Δ pex5 mutants or in other mutants defective in peroxisomal import."

Line 118. 'as a substrate'.

Line 286. 'on medium' NOT 'on media'.

Line 299. 'indicated' NOT 'indicted'.

The Legend to Figure S5 does not jive with the figure. Cell extracts were analyzed, not cell membranes. The authors must correct. Again 'indicated' NOT 'indicted' on line 310.

Line 319. 'pelleted at' NOT 'pelleted with'.

Line 346. 'foci decorating peroxisomes' NOT 'foci decorated with peroxisomes'.

Figures 5 a and b. What are the triangles pointing at?

Legend to Figure S14b does not jive with Figure S14b.

Reviewer #4 (Remarks to the Author):

Summary:

Stehlik et al. identify the phosphatase Ptc5 in *S. cerevisiae* as, surprisingly, having a PTS1 for peroxisome matrix import as well as a transmembrane domain and mitochondrial presequence. They observe that Ptc5 is dually targeted to mitochondria and peroxisomes, in a manner requiring the Ptc5 PTS1 and the peroxisomal import receptor Pex5. They also demonstrate that Ptc5 inserts its TMD into the mitochondrial inner membrane and requires cleavage by Imp1 at a specific defined site to liberate it to the IM space and allow its peroxisomal import. From this, they propose a novel 'tug-of-war' mechanism for trafficking of Ptc5 from mitochondria to peroxisomes, and suggest this exists to prevent Ptc5 from exerting its toxic effects in the cytosol. They also identify a possible peroxisomal function of Ptc5, dephosphorylating the peroxisomal enzyme Gpd1 to promote its activity. Importantly, they identify several other proteins with similar motifs that also exhibit dual mito/peroxisome targeting, suggesting this may be a more generic mechanism for protein trafficking between these organelles.

General comments:

Overall, this paper makes a number of interesting observations and comes up with a very intriguing and novel model for protein targeting to peroxisomes via mitochondria, which seems to be convincingly evidenced.

A weakness/criticism is that several of the really interesting observations are not very well explained or discussed in any great detail and are sometimes buried in the Figures. The

manuscript would benefit from being re-arranged, better explained and expanded with more detail and discussion to highlight its significance. Furthermore, important data from the Supplementary Figures needs to be added to the main Figures.

The manuscript could be more impactful if it were restructured/refocused. Some of the most interesting and important points, such as the fact that this targeting mechanism may be common to a number of proteins rather than more restricted to yeast Ptc5, are only addressed at the end. The authors may consider starting with the demonstration of dual peroxisome matrix/mitochondria targeting of a number of candidate proteins (Suppl. Figs. 13 and 14), and then focus on Ptc5 as one example to elucidate the mechanism. The paper would be strengthened if the mitochondria-peroxisome tug of war model could be shown to be more relevant, as well as under what conditions this might occur. In this respect, the authors may want to discuss the example of mammalian ACBD2 (Fan et al., *Mol Endocrinol.* 2016 Jul;30(7):763-82) for which a similar sort of idea was proposed, albeit less elegantly than shown here.

The focus appears unevenly spread – some of the really appealing functional data, e.g. the phosphatase function of Ptc5 in peroxisomes, how this mechanism might prevent its toxic effects in the cytosol, is rushed through at the end. I would be interested to see this expanded on and discussed more as this is an important and interesting aspect.

In line with this, the intriguing observation that mitochondria-peroxisome tug of war may take place at organelle membrane contact sites should be followed up and expanded experimentally, as this would increase novelty and impact.

Specific issues:

- The introduction would benefit from a more detailed examination (perhaps with examples) of known strategies for dual targeting, to highlight this is a novel mechanism.
- The authors assert that 'Pex5 appears to bind the C-terminus of Ptc5 during mitochondrial import'. Is this necessarily evident from the data presented, and are there any other possibilities to explain their findings? More discussion or experiments would help to clarify (see below).
- The observation that the tug-of-war mechanism can induce organelle contacts is very interesting, and the authors should expand on this. Does transit of Ptc5 from mitochondria to peroxisomes require membrane contact sites, or do the experimental manipulations lead to unspecific tethering/clustering of organelles? Fzo1 and Pex34 have been shown to tether peroxisomes and mitochondria in yeast (Shai et al., *Nat Commun.* 2018 May 2;9(1):1761), and the requirement of contact sites for the transit of Ptc5 could be tested in mutant strains.
- In line with this, the authors observed decoration of mitochondria with peroxisomes when expressing a Ptc5 variant lacking the transmembrane domain. Is there also increased peroxisome-mitochondria contact when the Ptc5 variant lacking the Imp1 cleavage site is expressed? Similarly, under conditions when Imp1 or Imp2 are deleted. If you now manipulate Ptc5 under those conditions (e.g. double Imp1 Ptc5 KO; manipulation of PTS1), would contacts be reduced? The authors should quantify those interactions. Alterations of interactions under those conditions would also support the proposed tug-of-war model.
- Figure 3E. The data imply that when the Ptc5 version lacking the PTS is expressed around 75% of peroxisomes are in proximity to mitochondria. So the majority of mitochondria is "decorated by a peroxisome", and this increases to approx. 85% when the Ptc5-PTS construct is expressed (?). How is this under normal conditions with no expression of Ptc5?
- Suppl. Figure 11: The expression of Sec63-mRFP-PTS_{Ptc5} appears to recruit almost all the peroxisomes in close proximity to the ER. Is this because the Sec63 cannot be cleaved (as Ptc5 is by Imp1), and are there any cellular consequences to this artificial tethering?
- Overall, there is little quantification of protein localisation to peroxisomes/mitochondria, and the authors should improve this throughout the study. Is Ptc5 only targeted to a subset of peroxisomes (as observed for mitochondria derived vesicles (MDVs) in mammalian cells)?
- More background to Ptc5 should be given as it is not clear what is known about it already – and some basic information should be provided, e.g. is anything known about its function at both

mitochondria and peroxisomes? Is it already known to be cleaved by Imp1? Why did previous reports not find it at peroxisomes if it is so prevalent there? What phenotype does a Ptc5 KO have? Is Ptc5 induced under oleate conditions or in the knockout does it impact growth when the cells are grown on oleate?

- Have the authors investigated an N-terminally tagged fusion of Ptc5? Does this block mitochondrial targeting; does the protein accumulate in the cytosol, but can be targeted (more slowly?) to peroxisomes? Does this impact on cell growth?
- The authors rule out a mistargeting effect of overexpressed Ptc5 by demonstrating that the endogenously tagged protein is also dual localised. As this is the most convincing data it should be moved to the main figure.
- The authors present the interesting hypothesis, that transient residence of Ptc5 in the cytoplasm has toxic effects so it is inserted into mitochondria before it folds and then transited directly to peroxisomes to prevent toxic effects. The authors should strengthen this intriguing hypothesis/concept. Are there other examples, e.g. for the ER or other organelles? Are there examples in higher eukaryotes, or is this specific to *S. cerevisiae*? The authors also identified other yeast proteins with dual mitochondria and peroxisome targeting (Dpi8, Mss2, Pxp2). Do they also contain an Imp1 cleavage site and are they targets for Imp1? Could this be an Imp1 specific mechanism? The authors may want to add the domain structure, PTS1 sites and details for these proteins. Are other Imp1 targets known? Please discuss.
- Ptc5-delta1-83 shows a modest growth defect on 2% glucose, more on ethanol, which is exacerbated when the PTS1 signal is removed. How about growth of this strain on oleate? As this would increase peroxisome numbers, and thus import of the protein. Does Ptc5-delta1-83 also dephosphorylate cytosolic Gpd1? As this could be what causes the toxicity?
- Can the authors explain why the RFP-PTS1 version of Rml2 is not targeted to peroxisomes?
- Is Ptc5 under control of its own endogenous promoter throughout?
- In a similar way does an Imp1 KO or a Ptc5 KO or a Gpd1 KO show growth defects on glucose or oleate? How about the double KOs? If Gpd1 KO shows defects and Ptc5 KO also shows defects, these experiments may support the model under physiological conditions; especially if experiments are not performed with endogenous levels of Ptc5 (see above).
- Would Fig. 1f fit better as part of Fig. 2, seeing as it concerns the Imp1 processing? Also, this figure does appear to show some colocalisation of Ptc5-RFP-PTS with a peroxisome marker in some Δ imp1 cells – can the authors explain this?
- Suppl. Fig 5b shows less Ptc5 (over)expression in the Δ imp1 cells – can the authors speculate why this is? Does this confound any of their interpretation about localisation?
- Do the authors know what happens to the TMD of Ptc5 that is left in the IMM following Imp1 processing?
- The authors state that 'Presumably, the detour via mitochondria is used to prevent premature folding'. Do they think this is the only function of this detour, or does Ptc5 have a function at mitochondria as well (e.g. dephosphorylation in IM space)?

Minor points:

Throughout, essential data is relegated to Supplementary Figures e.g. both Fig 1c and Suppl. Fig 1 are needed to demonstrate the peroxisome/mitochondria localisation of the overexpressed Ptc5 constructs, therefore this kind of data needs to be moved to the main Figures (e.g. Suppl. Figure 13 has a better format).

Throughout, there is little detail about the expression strategies used (e.g. endogenous vs overexpression, which promoters driving expression), and the rationale for using certain mutants that would be useful in interpreting the data. The authors may want to add a table/schematic of the localisation of the different mutants.

Throughout, the figure legends are significantly lacking in detail, making it difficult to follow the experiments. Molecular mass markers should be added to immunoblots to highlight migration differences.

Figure 2A-C are just an extension of Fig. 1 so could be re-organised.

Figure 3B. Some cells have no red signal? What is the reason for this? Please explain.

Figure 3C/D. The authors may want to use an independent marker to label mitochondria and peroxisomes to confirm that peroxisome-mitochondria proximity is altered. From the images shown it looks like there are fewer peroxisomes? Are peroxisome numbers altered under those conditions?

Please give more details on the Sec63 experiment.

The density gradient experiments in Suppl. Fig 3 and 5a need better labelling and discussion. Additional control/marker proteins should be added to show how reliable the separation is. In some gradients (Fig S3), Ptc5-RFP appears to have some similarity to Sps19 distribution(?).

Figure S5: what do the triangles under the blot indicate? In the Imp1 strain why is the migration of peroxisomal Ant1 also apparently altered in the gradient?

The authors should better explain the use of Δ coq9 cells. Similarly, why does truncating the C-terminus of Ptc5 help to discriminate between the unprocessed and mature forms? More explanation of the rationale behind the experiments is needed.

Ptc5 D302/D424 mutants are used in Fig. 3g – I presume these are the phosphatase-dead ones, but this is not mentioned anywhere. Similarly, D84K mutant in Suppl. Fig 8.

Fig S12. Why does peroxisomal/cytosolic Gpd1 migrate the same as mitochondrial porin in the gradient?

Figure S14 b suggests regulation of the tug-of-war mechanism as one cell shows only mitochondrial targeting and one peroxisome targeting. Is this a one off random event, or observed routinely? Why is Dpi8-RFP-PTS again added here (see Suppl. Figure 13)?

The authors address the function of Ptc5 at peroxisomes and nicely show that it dephosphorylates Gpd1 which they suggest increases Gpd1 enzyme activity so is an activator. What does this then mean with respect to cell physiology? When does Gpd1 need to be activated? Can this somehow be linked to IMP1 activation?

Suppl. Excel sheets: The authors may want to improve annotation, e.g. explain pI4w53. Is this a list of all the potential proteins with a mitochondrial targeting signal at the N-terminus and a C-terminal PTS1 in yeast? What does targeted mean? Have these all been tested? RML2 had a low "score" but was targeted to mitochondria (but not peroxisomes) based on the data presented (?). Please explain.

We thank all reviewers for their constructive comments, which helped a lot during the revision process.

Reviewers' comments:

Reviewer #1 (Remarks to the Author):

The report submitted by Stehlik et al. studies a mechanism for the targeting of a subset of proteins to peroxisomes following their translocation into the mitochondria. The main body of their work focuses on Ptc5, a protein phosphatase originally described as localising to the intermembrane space of the mitochondria, whereby they bioinformatically identify a putative type 1 peroxisomal targeting signal at its C-terminus. Through a set of imaging and biochemical experiments, using a range of truncation constructs, they propose the following model: Ptc5 is first imported into the mitochondria via its N-terminal targeting sequence, where it is cleaved by the IMP machinery between amino acids 83 and 84. Ptc5 can then associate with the peroxisomal import machinery (e.g. Pex5) to then translocate from the mitochondria to the peroxisomal lumen. In doing so, the deleterious effects of Ptc5 phosphatase activity residing in the cytoplasm can be avoided. The experiments supporting this model are well controlled and, overall, convincing. Their work adds further weight to the developing idea that there is a complex interplay between these two metabolic organelles.

The authors have provided a broad range of data in support of their model that Ptc5 can localise to both mitochondria and peroxisomes. However, whilst their data showing that their constructs can localise to both organelles are compelling, I'm a little worried that they don't definitively show that this would indeed occur with the wild-type protein.

Main points

1. Most of the evidence stems from the fact that their reporter constructs show evidence of cleavage by the IMP machinery in the mitochondria while accumulating in peroxisomes. Yet, it would be important to discount the possibility that some of the IMP peptidase resides in the peroxisome.

We approached this point by two different experiments. To follow the localization of the IMP machinery we endogenously myc-tagged Imp2. We show by density gradient centrifugation that the biologically active Imp2-3xmyc variant is only visible in mitochondrial fractions (Fig. 4). In addition, we constructed a variant of Ptc5, which lacks the mitochondrial presequence but retains the transmembrane domain and the Imp1 cleavage site. We show that this protein fused to RFP was imported into peroxisomes but was not processed (Fig. 4). Thus, we are convinced that IMP dependent processing of Ptc5 occurs only inside mitochondria.

2. Much of the data relies on variants of the Ptc5-RFP-PTS construct. It is important to ensure that the RFP is not detrimental to full translocation into the mitochondria, allowing more time for the PTS1 to have a role in the localisation of the protein. The Ptc5-3xHA-PTS expressing yeast certainly go a long way in testing this possibility, as the protein is endogenously expressed and only has a small tag; however, from the representative images, this construct is far more localized to mitochondria than the RFP construct. This raises the worrisome possibility that much of the peroxisomal signal observed here is artifactual and due to the PTS lingering longer in the cytosol due to the time necessary to unfold and translocate the RFP. One way to test this is to quantify the amount of Ptc5-3xHA-PTS in the peroxisomal fraction and mitochondrial fractions.

The reviewer raises an important point. Indeed, the fraction of peroxisomal Ptc5-RFP-PTS determined by Western blotting was higher if compared to endogenously expressed Ptc5-3xHA-PTS (30% and 10%, respectively; see attached Western Blot). To improve immunofluorescence, we substituted 3xHA with 3xMyc, since this tag allows longer fixation (see Fig. 2). The immunofluorescence looked much better and also allowed for quantification. The ratio of peroxisomal/mitochondrial protein determined by Western blots derived from density gradient centrifugation were similar for both 3xMyc and 3xHA tagged proteins (Fig. 2c and Western blot below). Moreover untagged Ptc5 has been shown to physically interact with Pex14 in a recent study (Wroblewska et al., 2017; BBA). Together these data confirm peroxisomal localization of Ptc5.

3. There are a couple of issues with the data studying a functional role for Ptc5 at peroxisomes. The band shift in Supp. Fig. 12b doesn't necessarily show hyperphosphorylation of Gpd1 in the $\Delta ptc5$ cells. The addition of a λ -phosphatase treated sample and the inclusion of a "normal" SDS-PAGE blot (alongside the PhosTag gel) would control for many of the other potential causes of the band-shift observed. Furthermore, Supp. Fig. 12a is a little unclear. Currently it shows a massive enrichment of Gpd1 in the Por1-positive (i.e. mitochondrial) fractions, but the authors state it is localised to peroxisomes and cytoplasm. Both mitochondrial and peroxisomal markers might help to clear this up, as well as an indication of the lanes chosen for the PhosTag gel analysis.

We have now included a λ -phosphatase treated sample that confirms phosphorylation of peroxisomal Gpd1 in $\Delta ptc5$ mutants. We also provide another density gradient experiments with washed organelle pellets to remove cytosolic contamination, which is also found at the top of the gradient. For PhosTag analysis Gpd1-GFP of dense peroxisomal fractions was analyzed (Fig. 5f and Supplementary Fig. 6a).

4. Finally, an alternative model for the enzyme-dead mutants rescuing the growth defect observed in Fig. 3h-g is that these mutants are not expressed well, or that they no longer localise to the cytoplasm.

We present western blot and microscopy data showing that expression levels of enzyme-dead versions are even higher than the levels of the progenitor protein (Fig. 5j). Localization of the mutated fusion proteins was not altered (Supplementary Fig. 6b).

Minor points

1. It is concluded that Ptc5 Δ TM-RFP-PTS can induce contact between mitochondria and peroxisomes. Using confocal microscopy, it is not possible to conclude that the organelles are in fact contacting one another. As a result, the language would need to be softened.

The language was softened.

2. Whilst the data on the dual localisation of Ptc5 are convincing, it isn't clear from the images provided that this is a general phenomenon for a subset of proteins (in this case: Pxp2, Mss2, Dpi8 and Rml2). Arguably, the images presented in Supp. Fig. 13&14 only show dual targeting for Dpi8, but the authors conclude that 3 out of 4 candidates showed this phenotype.

We now provide additional evidence for dual targeting of Pxp2-RFP-PTS1 to mitochondria and peroxisomes. Specifically, we show density gradient experiments and study localization of the RFP fusion proteins. We found that Pxp2-RFP co-migrates with Por1. Thus foci observed for this protein probably represent mitochondrial sub-compartments (Fig. 6a and 6c). We also show that the N-terminus of Pxp2 contains a functional mitochondrial presequence (Fig. 6d). Furthermore, we demonstrate that expression of Pxp2-RFP-PTS1, Mss2-RFP-PTS and Dpi8-RFP-PTS increases the number of peroxisomes associated with mitochondria (Fig. 6e and Supplementary Fig. 7; similar to Ptc5^{ATM}-RFP-PTS1). Therefore, we discuss molecular tug-of-war as a potential mechanism to enhance the association of peroxisomes and mitochondria in greater detail in the revised version.

3. In the final remarks of the paper, the authors state that there is other work highlighting a close relationship between peroxisomes and mitochondria, using Sugiura et al. (2017) to support their claim. The work shown in the referenced paper is based on a phenomenon that occurs in mammalian cells, but is not observed in yeast, i.e. PEX3 localising to mitochondria (indeed peroxisomal biogenesis is quite different between yeast and mammalian cells). Therefore, this is perhaps not the best evidence to support their point; certainly not on its own. Their claim, that there is a close relationship between the two compartments, arguably has substantial evidence in support of it, e.g. overlap in metabolic functions (oxidation of fatty acids, lipid synthesis, ROS metabolism); many proteins are shared – Fis1 (also shown in yeast), MUL1/MAPL, USP30, Miro1/2 (also shown in yeast, known as Gem1) etc.; they share the fission machinery. If true, the work shown in this manuscript is an important addition to this, as it shows that some proteins can transit through the mitochondria first. It might, therefore, be worth more comprehensively discussing this idea.

We appreciate this point and have adapted the discussion section accordingly.

4. A schematic for the constructs generated, as well as stating the specific amino acid changes, would be useful.

We added a Supplementary Figure providing a scheme for many of constructs used in this study. We also highlight the subcellular localization of respective fusion proteins in this figure.

Reviewer #2 (Remarks to the Author):

In this manuscript, Stehlik et al. report a new protein sorting pathway connecting mitochondria and peroxisomes. Specifically, the authors have identified several proteins that are targeted to the mitochondrial intermembrane space, processed by IMP (the mitochondrial inner membrane peptidase) and, finally, re-targeted to the peroxisome. Such a pathway is new and thus of considerable interest to all molecular/cellular biologists working on mitochondria and/or peroxisomes. With a few exceptions (see below), the data are solid. I have some questions and suggestions.

Major issue:

The data described in this work for Ptc5-RFP-PTS are rather interesting per se, and, as stated above, unveil a protein sorting pathway that was never described before. However, there may be some doubts regarding the biological relevance of these findings because the RFP moiety in the fusion protein may fold immediately after synthesis in the cytosol leading to an artificial arrest of the protein at the mitochondrial import sites (see for instance Rassow J et al. (1989) JCB 109:1421). It is highly unlikely that the same happens with the Ptc5-3HA-PTS protein. Therefore, the results obtained with this protein should be presented and discussed in the main body of the manuscript and not in supplementary fig 4.

Thanks for this advice. We have now included a main figure showing this data. Note that we now used a 3xMyc tag instead of the 3xHA tag to improve the quality of the immunofluorescence microscopy (longer fixation possible). We show that 3xMyc and 3xHA tagged variants behave similar in density gradient centrifugation experiments (see Western blot below and Western blot in Fig. 2).

Other issues:

Line 50 – reference to the experiment shown in Supplementary fig 1 – the mitochondrial localization of Ptc5-RFP-PTS (3 right panels) is hard to see. Please improve image quality

We added a picture with enhanced signal to show mitochondrial staining.

Line 62 – “...abolished sorting of Ptc5-RFP-PTS1 into peroxisomes”. There are a few “yellow” dots in Fig.1F, which I presume represent red and green labelling in different planes. If so, this should be acknowledged and explained.

The reviewer is correct. Therefore, we softened the language to “... strongly reduced sorting of Ptc5-RFP-PTS1 to peroxisomes”. We did this because there is also incidental co-localization of peroxisomes and mitochondria at the resolution of a usual microscope. Thus, residual peroxisomal targeting cannot be formally excluded.

Lines 62-63 – “Supplementary figure 5a”- I am not sure the Nycodenz gradients are really of help here. It is very difficult (if not impossible) to get a good separation between yeast mitochondria and peroxisomes. The same happens here. Thus, the experiment is not really conclusive (in the delta-imp1 strain, there is a small pool of Ptc5-RFP-PTS at the bottom of the gradient together with peroxisomes). Consider removing these data. Also, note that the triangles in the figure are not explained and “Ptc5-RFP” in the upper panel should be “Ptc5-RFP-PTS”

We agree that the data is not as solid as the data presented in Figure 1. Organelles were prepared from glucose grown cells ($\Delta imp1$ cells do not grow on non-fermentable carbon sources such as oleic acid). At least in our hands, this hampers separation between yeast mitochondria and peroxisomes. We nevertheless decided to keep the figure, since enrichment of Ptc5-RFP-PTS compared to Ptc5-RFP is visible in the dense fractions. We corrected the mistake you mentioned. Triangles represent fractions analyzed for processing by Imp1/migration in SDS-PAGE (Fig. 3).

Line 70- “(supplementary Fig. 6)”- These data are very important and this experiment should be done correctly. At pH 11.5 (0,1 M Na₂CO₃) organelle membranes are disrupted yielding membrane fragments of low density (see Fujiki Y et al. (1982) JCB 93:97). Thus, they are recovered by ultracentrifugation (100.000xg, 1 h.). A 13k spin for 10 min is really not enough and this is the reason why in the delta-imp1 strain significant amounts of both Ptc5-RFP-PTS and Por1 appear in the Na₂CO₃ S fraction.

This was a very valuable suggestion. We repeated the experiment with a 100k spin and indeed ended up with better data (Supplementary Fig. 3).

Line 73 – the Deltacoq9 cells. Please provide a sentence to explain why these cells have a mitochondrial respiratory defect.

We have added a short explanation.

Line 82 – “supplementary table 1” – Maybe I missed it, but I could find no data in this table showing mapping of the IMP1 cleavage site on the Ptc5 protein.

The cleavage site is visible from the data presented in the coverage section. Comparison of peptides derived from band 2 representing the processed Ptc5 variant and those derived from band 3 from $\Delta imp1$ cells demonstrates that processing takes place at LSL/D.

Lines 105-109 – It seems that the authors are assuming that despite lacking a TM domain, this Ptc5 protein still localizes to the mitochondrial intermembrane space. Is this so? Shouldn't this protein end up in the mitochondrial matrix? Related to this issue: did the authors try protease-protection assays to show that mitochondrial Ptc5 proteins expose their C-termini into the cytosol?

We did not try protease protection assays. As the reviewer correctly states, the Ptc5 version lacking the TM domain should be sorted to the mitochondrial matrix instead of the IMS. We detected Ptc5^{ΔTM}-RFP-PTS1 signal in peroxisomes not associated with mitochondria, indicating that translocation is possible albeit with lower efficiency. Since Ptc5^{ΔTM}-RFP-PTS1 is an artificial construct directed to the mitochondrial matrix, we wondered if native proteins containing a mitochondrial targeting signal and a PTS1 might be sorted the same way. This resulted in the idea to search for mitochondrial proteins harboring a PTS1 as it indicates that tug-of-war like sorting is also possible for mitochondrial proteins that do not follow the IMP pathway. These proteins might end up either in the peroxisomal or in the mitochondrial matrix. However, we assume that a transient translocation intermediate connected to both organelles similar to that of Ptc5 exists.

Line 122- “hyperphosphorylated”. Why not just “phosphorylated”? Is a reference missing?

“Hyperphosphorylated” was changed to phosphorylated.

Lines 137-138 – The authors should provide a bit more information regarding the proteins shown in Fig. S13 and Fig. S14, and the results obtained (just as they did for Dpi8 in legend to Fig. S14).

We added information in the main text (lines 208-228).

Line 310 – “a, Membrane of indicted...”. The authors mean “a, Organelles of indicated...”, right? If not, I saw no reference to the preparation of membranes.

You are right. The sentence was changed accordingly.

Line 458- A reference for the method is missing.

The reference was added.

Supplementary fig 3 – It would be easier to understand this figure if the “top” and “bottom” of the gradients

were indicated in the figure. Also, the small pool of Ptc5-RFP co-sedimenting with peroxisomes (GFP-Sps19) probably represents some mixed mitochondria/peroxisome aggregates (Por1 is also visible in these fractions). A small comment on this pool would also help to understand the results.

We rearranged the pictures of the gradient experiment according to your suggestion.

Reviewer #3 (Remarks to the Author):

In this manuscript, Stehlik and colleagues provide evidence that the type 2C protein phosphatase Ptc5 of yeast is dually localized between mitochondria and peroxisomes. Combining genetic, microscopic and biochemical methodologies, the authors provide strong initial, but not yet conclusive, evidence that Ptc5 is targeted first to the mitochondrial intermembrane space (IMS) by an N-terminal mitochondrial presequence that is cleaved in the IMS, and that some of the cleaved Ptc5 still in the mitochondrial translocon, where it remains linear in structure, is then pulled into the peroxisome via the activity of the peroxisomal targeting signal 1 (PTS1) receptor Pex5 that interacts with Ptc5's degenerate PTS1 (-PRL) at its carboxyl terminus. This all leads to import of Ptc5 into the peroxisomal matrix.

Although the initial evidence for the pathway of targeting of Ptc5 first to the mitochondrion and then to the peroxisome via the activity of Pex5 is strong, it is not conclusive. The authors have to:

1) provide time-course evidence, either by biochemical or microscopic pulse-chase analysis, that Ptc5 is first targeted to the mitochondrion and then directed to the peroxisomal matrix.

There is evidence that proteins carrying both a mitochondrial and a peroxisomal targeting signal are predominantly imported into mitochondria (e.g. Szewczyk et al. 2001; JBC) most likely due to the fact that mitochondrial import could occur co-translationally (e.g. Danpure 1997; Bioassays). We have shown that loss of IMP processing interferes with peroxisomal import of Ptc5-RFP-PTS1 (Fig. 3a). We now also present evidence that the IMP machinery is only present in mitochondria (Fig. 4a). Moreover, we show that a Ptc5 derivative lacking the presequence is targeted to peroxisomes but not processed by the IMP complex (Fig. 4c). Since peroxisomal Ptc5 is entirely processed (as shown for example in Figure 3c and 3d), at least the N-terminus must have entered mitochondria prior to its translocation to the peroxisomal matrix. Otherwise it won't be processed. The exact timing of these interactions was not specifically addressed, as they are likely to occur on a short time scale. We excluded retro-translocation of Ptc5, when Pex5 was absent (e.g. Fig 1c). Therefore the interactions probably even occur simultaneously.

2) provide protein-protein interaction evidence (e.g. pull-downs) to show that Ptc5 directly interacts through its degenerate PTS1 with the PTS1 receptor Pex5.

We performed a yeast two-hybrid assay to demonstrate interaction of the PTS1 of Ptc5 with Pex5 (Supplementary Fig. 1c).

Additional points the authors should address are:

Line 68. 'reduced mobility' NOT 'lower mobility'.

Line 82. 'These data were confirmed' NOT 'This data was confirmed'.

Lines 85-86. The data only 'suggest' but do not conclusively 'show' that "Ptc5 is first targeted to mitochondria, processed by the IMP machinery and then sorted to peroxisomes." (See Point 1 above.)

Lines 92-93. Cumbersome sentence structure. Rephrase as 'We failed to observe cytosolic localization of Ptc5-RFP-PTS either in Δ pex5 mutants or in other mutants defective in peroxisomal import.'

Line 118. 'as a substrate'.

Line 286. 'on medium' NOT 'on media'.

Line 299. 'indicated' NOT 'indicted'.

The Legend to Figure S5 does not jive with the figure. Cell extracts were analyzed, not cell membranes. The authors must correct. Again 'indicated' NOT 'indicted' on line 310.

The legend was corrected.

Line 319. 'pelleted at' NOT 'pelleted with'.

Line 346. 'foci decorating peroxisomes' NOT 'foci decorated with peroxisomes'.

Figures 5 a and b. What are the triangles pointing at?

This data is now in Fig. S2. Triangles denote fractions analyzed by high resolution SDS-PAGE.

Legend to Figure S14b does not jive with Figure S14b.

We have corrected the other points mentioned above.

Reviewer #4 (Remarks to the Author):

Summary:

Stehlik et al. identify the phosphatase Ptc5 in S. cerevisiae as, surprisingly, having a PTS1 for peroxisome matrix import as well as a transmembrane domain and mitochondrial presequence. They observe that Ptc5 is dually targeted to mitochondria and peroxisomes, in a manner requiring the Ptc5 PTS1 and the peroxisomal import receptor Pex5. They also demonstrate that Ptc5 inserts its TMD into the mitochondrial inner membrane and requires cleavage by Imp1 at a specific defined site to liberate it to the IM space and allow its peroxisomal import. From this, they propose a novel 'tug-of-war' mechanism for trafficking of Ptc5 from mitochondria to peroxisomes, and suggest this exists to prevent Ptc5 from exerting its toxic effects in the cytosol. They also identify a possible peroxisomal function of Ptc5, dephosphorylating the peroxisomal enzyme Gpd1 to promote its activity. Importantly, they identify several other proteins with similar motifs that also exhibit dual mito/peroxisome targeting, suggesting this may be a more generic mechanism for protein trafficking between these organelles.

General comments:

Overall, this paper makes a number of interesting observations and comes up with a very intriguing and novel model for protein targeting to peroxisomes via mitochondria, which seems to be convincingly evidenced. A weakness/criticism is that several of the really interesting observations are not very well explained or discussed in any great detail and are sometimes buried in the Figures. The manuscript would benefit from being re-arranged, better explained and expanded with more detail and discussion to highlight its significance. Furthermore, important data from the Supplementary Figures needs to be added to the main Figures.

We now moved data from the Supplementary Figures to the main figures (e. g. data about the other candidates, which are no substrates of the IMP complex but may be transported via molecular tug-of-war). We provide additional data for these proteins. Specifically, we show by density gradient centrifugation that foci detected for Pxp2-RFP are located inside mitochondria and may represent mitochondrial subdomains. We also show that expression of Pxp2-RFP-PTS, Mss2-RFP-PTS and Dpi8-RFP-PTS increases the number of peroxisomes decorating mitochondria. We added a paragraph with an improved explanation of the data.

The manuscript could be more impactful if it were restructured/refocused. Some of the most interesting and important points, such as the fact that this targeting mechanism may be common to a number of proteins rather than more restricted to yeast Ptc5, are only addressed at the end. The authors may consider starting with the demonstration of dual peroxisome matrix/mitochondria targeting of a number of candidate proteins (Suppl. Figs. 13 and 14), and then focus on Ptc5 as one example to elucidate the mechanism. The paper would be strengthened if the mitochondria-peroxisome tug of war model could be shown to be more relevant, as well as under what conditions this might occur. In this respect, the authors may want to discuss the example of mammalian ACBD2 (Fan et al., Mol Endocrinol. 2016 Jul;30(7):763-82) for which a similar sort of idea was proposed, albeit less elegantly than shown here.

We thank the reviewer for these suggestions. We agree that this rearrangement could lead to a different perception of our story. However, we decided to not completely rearrange our manuscript, since we did most of the work on Ptc5. The major message of this paper is the discovery of a novel sorting pathway from mitochondria to peroxisomes and its biological significance. According to our data Ptc5 is the only PTS1 containing protein, which is an Imp1 substrate in yeast. The other candidates are likely to contain a usual mitochondrial presequence and a PTS1 but are not IMP substrates (as now shown for Pxp2; see above). We became interested in these proteins while analyzing Ptc5-RFP-PTS lacking the trans-membrane domain. Although they are probably transported via a tug-of-war like mechanism, the underlying molecular biology is a little different. We added the suggested reference in our discussion section, as it indicates that tug-of-war like sorting is evolutionary conserved (many thanks for this advice).

The focus appears unevenly spread – some of the really appealing functional data, e.g. the phosphatase function of Ptc5 in peroxisomes, how this mechanism might prevent its toxic effects in the cytosol, is rushed through at

the end. I would be interested to see this expanded on and discussed more as this is an important and interesting aspect.

We now added headlines to every paragraph, which hopefully clarifies the significance of the data. We also added an experiment which shows that the phosphatase-dead variants are expressed in even higher amounts compared to the wildtype version. This supports our claim that mislocalized phosphatase activity is responsible for the detected growth phenotype. We also added the expanded results of PhosTag experiments in a main figure (Fig. 5f and 5g).

In line with this, the intriguing observation that mitochondria-peroxisome tug of war may take place at organelle membrane contact sites should be followed up and expanded experimentally, as this would increase novelty and impact.

We agree that this is a very interesting point. However, we think that a comprehensive investigation of the contacts is outside the scope of the current manuscript but may lead to a different paper, with a focus on tug-of-war and organelle contacts. Still we provide novel data for Pxp2, Mss2 and Dpi8 as stated above.

Specific issues:

- *The introduction would benefit from a more detailed examination (perhaps with examples) of known strategies for dual targeting, to highlight this is a novel mechanism.*

We did not specifically discuss more examples in greater detail. We nevertheless think that the suggestion is good. Thus, we highlight the differences between Ptc5 sorting and already characterized mechanisms for dual targeting in our discussion section, which is now added to the manuscript.

- *The authors assert that ‘Pex5 appears to bind the C-terminus of Ptc5 during mitochondrial import’. Is this necessarily evident from the data presented, and are there any other possibilities to explain their findings? More discussion or experiments would help to clarify (see below).*

We do not observe retro-translocation of Ptc5 to the cytosol in $\Delta pex5$ cells as stated in the text. In addition, we show that Ptc5 is stable in the cytosol. Therefore, we consider binding of Ptc5 to Pex5 prior to complete import as the most likely mechanism (also stated in the text).

- *The observation that the tug-of-war mechanism can induce organelle contacts is very interesting, and the authors should expand on this. Does transit of Ptc5 from mitochondria to peroxisomes require membrane contact sites, or do the experimental manipulations lead to unspecific tethering/clustering of organelles? Fzo1 and Pex34 have been shown to tether peroxisomes and mitochondria in yeast (Shai et al., Nat Commun. 2018 May 2;9(1):1761), and the requirement of contact sites for the transit of Ptc5 could be tested in mutant strains.*

This is a reasonable idea. We have tested an influence of both $\Delta fzo1$ and $\Delta pex34$ mutants on Ptc5-RFP-PTS1 translocation *in vivo*. We did not include the data since we didn't see significant differences. This is in agreement with an assumption made by Shai et al. 2018. Tethers can be functionally redundant and thereby escape detection in a screen based on libraries with single deletion mutants. Molecular tug-of-war even could explain this redundancy as now stated in the discussion section (lines 252-259).

- *In line with this, the authors observed decoration of mitochondria with peroxisomes when expressing a Ptc5 variant lacking the transmembrane domain. Is there also increased peroxisome-mitochondria contact when the Ptc5 variant lacking the Imp1 cleavage site is expressed? Similarly, under conditions when Imp1 or Imp2 are deleted. If you now manipulate Ptc5 under those conditions (e.g. double Imp1 Ptc5 KO; manipulation of PTS1), would contacts be reduced? The authors should quantify those interactions. Alterations of interactions under those conditions would also support the proposed tug-of-war model.*

This was an interesting suggestion. We did not observe a significant increase of peroxisomes associated with mitochondria in *imp1* or *imp2* mutants or for the uncleavable Ptc5^{ALSLD}-RFP-PTS. This may result from the altered morphology of mitochondria observed in these strains. However, we find that Pxp2-RFP-PTS1, Mss2-RFP-PTS and Dpi8-RFP-PTS, which probably contain a usual mitochondrial presequence (shown for Pxp2 in Fig. 6) enhance the association of both organelles (Fig. 6e and Supplementary Fig. 7). These fusion proteins are comparable to the ΔTM version of Ptc5, since none of them can be laterally released into the inner mitochondrial membrane but are either pulled into the mitochondrial matrix or the peroxisomal lumen. This mechanism is more likely to increase contact between both organelles because more force can be generated inside mitochondria.

• *Figure 3E. The data imply that when the Ptc5 version lacking the PTS is expressed around 75% of peroxisomes are in proximity to mitochondria. So the majority of mitochondria is “decorated by a peroxisome”, and this increases to approx. 85% when the Ptc5-PTS construct is expressed (?). How is this under normal conditions with no expression of Ptc5?*

We now added data for Pxp2-RFP-PTS, Mss2-RFP-PTS and Dpi8-RFP-PTS, which were also increasing the number of peroxisomes in proximity to mitochondria. As control we always used strains expressing respective fusion proteins without C-terminal PTS1. The number of peroxisomes associated with mitochondria ranged from 60% – 70% in control strains, but the increased association in strains expressing PTS containing variants was significant. Other labs already came up with similar data of peroxisomes associated with mitochondria in mammalian cells (around 60%; Valm *et al.* 2017, Nature).

• *Suppl. Figure 11: The expression of Sec63-mRFP-PTS_{Ptc5} appears to recruit almost all the peroxisomes in close proximity to the ER. Is this because the Sec63 cannot be cleaved (as Ptc5 is by Imp1), and are there any cellular consequences to this artificial tethering?*

Pex5 has continuous access to the PTS1 of Sec63. This may be the reason for the efficient tethering observed. Expression of this construct probably alters cellular physiology. However, we were not sure what meaning the other outcome of testing this would have.

• *Overall, there is little quantification of protein localisation to peroxisomes/mitochondria, and the authors should improve this throughout the study. Is Ptc5 only targeted to a subset of peroxisomes (as observed for mitochondria derived vesicles (MDVs) in mammalian cells)?*

We did not observe a significant bias towards a subpopulation of peroxisomes. We added quantifications for a number of the experiments.

• *More background to Ptc5 should be given as it is not clear what is known about it already – and some basic information should be provided, e.g. is anything known about its function at both mitochondria and peroxisomes? Is it already known to be cleaved by Imp1? Why did previous reports not find it at peroxisomes if it is so prevalent there? What phenotype does a Ptc5 KO have? Is Ptc5 induced under oleate conditions or in the knockout does it impact growth when the cells are grown on oleate?*

This is a fair point. We now include additional information about Ptc5. We did not observe a growth defect for the *ptc5* mutant on linolic acid containing plates. In a previous proteomics approach Ptc5 was shown to co-purify with the peroxisomal membrane protein Pex14, but the authors did not follow up on this. Ptc5 was also suggested to regulate the activity of pyruvate dehydrogenase in the mitochondrial matrix. However, this could not be confirmed by later studies. The references are given in the manuscript now. Ptc5 was already known to be one of the IMP targets as also stated in the text.

• *Have the authors investigated an N-terminally tagged fusion of Ptc5? Does this block mitochondrial targeting; does the protein accumulate in the cytosol, but can be targeted (more slowly?) to peroxisomes? Does this impact on cell growth?*

N-terminal tagging of Ptc5 has been tried by another lab (Nötzel *et al.*, 2016, Traffic) and also by us. It does not result in a detectable signal.

• *The authors rule out a mistargeting effect of overexpressed Ptc5 by demonstrating that the endogenously tagged protein is also dual localised. As this is the most convincing data it should be moved to the main figure.*

We appreciate this comment, which was also brought up by other referees. We therefore performed experiments to strengthen peroxisomal targeting of the endogenous protein. Results are depicted in the novel figure 2. These data clearly indicate dual targeting of endogenously expressed Ptc5.

• *The authors present the interesting hypothesis, that transient residence of Ptc5 in the cytoplasm has toxic effects so it is inserted into mitochondria before it folds and then transited directly to peroxisomes to prevent toxic effects. The authors should strengthen this intriguing hypothesis/concept. Are there other examples, e.g. for the ER or other organelles? Are there examples in higher eukaryotes, or is this specific to *S. cerevisiae*? The authors also identified other yeast proteins with dual mitochondria and peroxisome targeting (Dpi8, Mss2, Pxp2). Do they also contain an Imp1 cleavage site and are they targets for Imp1? Could this be an Imp1 specific*

mechanism? The authors may want to add the domain structure, PTS1 sites and details for these proteins. Are other Imp1 targets known? Please discuss.

We are not aware of other examples for this concept yet, but we might have missed something in the literature. We now present novel data showing that it is not the expression level of the phosphatase dead variants but most probably the lack of activity of the protein, which is responsible for the phenotype. These data support our assumption. As stated above, the other proteins do not contain Imp1 cleavage sites. This is now discussed more specifically in the revised manuscript.

- *Ptc5-delta1-83 shows a modest growth defect on 2% glucose, more on ethanol, which is exacerbated when the PTS1 signal is removed. How about growth of this strain on oleate? As this would increase peroxisome numbers, and thus import of the protein. Does Ptc5-delta1-83 also dephosphorylate cytosolic Gpd1? As this could be what causes the toxicity?*

We tested this idea by expressing Ptc5 Δ 83-RFP in Δ gpd1 cells but did not find a difference. Thus, the toxic effect must originate from dephosphorylation of other proteins in the cytosol. We did not test growth on oleic acid as the strains already show a growth defect on the non-fermentable carbon source ethanol and are unlikely to grow on oleic acid. It is not clear yet if import of PTS proteins is faster in oleic acid as there is also more competition for Pex5.

- *Can the authors explain why the RFP-PTS1 version of Rml2 is not targeted to peroxisomes?*

Probably the PTS1 is not recognized or not accessible to Pex5. It also has a relatively low prediction value.

- *Is Ptc5 under control of its own endogenous promoter throughout?*

We expressed the RFP-tagged versions of Ptc5 under control of the *tef1* promoter to enhance its detection. The protein is still quite unstable. However, we observed dual targeting and processing of Ptc5 also for the endogenously expressed variant with a small tag (see Fig. 2).

- *In a similar way does an Imp1 KO or a Ptc5 KO or a Gpd1 KO show growth defects on glucose or oleate? How about the double KOs? If Gpd1 KO shows defects and Ptc5 KO also shows defects, these experiments may support the model under physiological conditions; especially if experiments are not performed with endogenous levels of Ptc5 (see above).*

Neither Δ ptc5 nor Δ gpd1 have a remarkable growth defect on glucose or oleate. Δ imp1 cells exhibit a growth defect on non-fermentable carbon sources such as ethanol or oleate. Therefore, these data neither support nor infer with our model.

- *Would Fig. 1 fit better as part of Fig. 2, seeing as it concerns the Imp1 processing? Also, this figure does appear to show some colocalisation of Ptc5-RFP-PTS with a peroxisome marker in some Δ imp1 cells – can the authors explain this?*

We revised the figures also because of novel data concerning endogenously tagged and expressed Ptc5. All the data derived from *imp* mutants is now depicted in Figures 3 and 4. Residual co-localization between mitochondria and peroxisomes is always observed. This could be caused by their tight association (see above) but also by collapsing z-stacks.

- *Suppl. Fig 5b shows less Ptc5 (over)expression in the Δ imp1 cells – can the authors speculate why this is? Does this confound any of their interpretation about localisation?*

This effect is not specific for the PTS1 containing fusion proteins and does not confound our interpretation. Ptc5 stuck in the inner mitochondrial membrane may be degraded faster because it is probably recognized as aberrant.

- *Do the authors know what happens to the TMD of Ptc5 that is left in the IMM following Imp1 processing?*

To our knowledge this has not been studied yet.

- *The authors state that ‘Presumably, the detour via mitochondria is used to prevent premature folding’. Do they think this is the only function of this detour, or does Ptc5 have a function at mitochondria as well (e.g. dephosphorylation in IM space)?*

Some data indicated that Ptc5 is the phosphatase for the pyruvate dehydrogenase subunit Pda1. This could not be confirmed by phospho-proteomics and instead Ptc6 was shown to dephosphorylate Pda1. These proteomics experiments identified Gpd1 (as our data also demonstrates) and a couple of mitochondrial proteins from the intermembrane space as potential targets of Ptc5. No one followed up on these proteins yet.

Minor points:

Throughout, essential data is relegated to Supplementary Figures e.g. both Fig 1c and Suppl. Fig 1 are needed to demonstrate the peroxisome/mitochondria localisation of the overexpressed Ptc5 constructs, therefore this kind of data needs to be moved to the main Figures (e.g. Suppl. Figure 13 has a better format).

Several of the points raised by the referee may have resulted from our manuscript being prepared as a short report. Due to the guidelines of *Nature communications* we have now prepared it as an article, which allows for more detailed explanation and more figures.

Throughout, there is little detail about the expression strategies used (e.g. endogenous vs. overexpression, which promoters driving expression), and the rationale for using certain mutants that would be useful in interpreting the data. The authors may want to add a table/schematic of the localisation of the different mutants.

See above; we added a supporting table with the constructs and their localization.

Throughout, the figure legends are significantly lacking in detail, making it difficult to follow the experiments.

We improved several of the figure legends to make it easier to follow.

Molecular mass markers should be added to immunoblots to highlight migration differences.

We did not include markers in the main figures. However, we now provide Western blots as original images together with other original data in a supporting file. In this file markers were included for the blots addressing the processing of Ptc5.

Figure 2A-C are just an extension of Fig. 1 so could be re-organised.

These figures are now reorganized.

Figure 3B. Some cells have no red signal? What is the reason for this? Please explain.

This is typical for low copy number plasmid driven expression in yeast. Plasmids may get lost even under selective conditions.

Figure 3C/D. The authors may want to use an independent marker to label mitochondria and peroxisomes to confirm that peroxisome-mitochondria proximity is altered. From the images shown it looks like there are fewer peroxisomes? Are peroxisome numbers altered under those conditions?

This experiment was based on Tim50-CFP and Ant1-YFP (compare to Pxp2 etc.). This is now specifically stated in the manuscript and in the figure legends. The number of peroxisomes was not significantly altered.

Please give more details on the Sec63 experiment.

We now specify that Sec63 is a membrane bound ER resident protein.

The density gradient experiments in Suppl. Fig 3 and 5a need better labelling and discussion. Additional control/marker proteins should be added to show how reliable the separation is. In some gradients (Fig S3), Ptc5-RFP appears to have some similarity to Sps19 distribution(?).

We are confident that the density gradient experiments shown are conclusive. A better resolution of peroxisomes and mitochondria is hard to achieve with yeast organelles. We added additional data from other experiments, quantified several Western blots and also provide novel density gradient experiments for other proteins to support our claims.

Figure S5: what do the triangles under the blot indicate? In the Imp1 strain why is the migration of peroxisomal Ant1 also apparently altered in the gradient?

The triangles denote fractions analyzed by high resolution SDS-PAGE (Fig. 3b). This is now stated in the legend. The gradient analysis for this experiment was a little tricky. *Δimp1* mutants do not grow on oleic acid (non-fermentable carbon source). Therefore peroxisome proliferation cannot be induced. This hampers a better separation. Please note that we used the clearest fractions possible for the experiment presented in Fig. 3b. Since the protein is entirely processed in WT cells, the message is even true without the gradient data.

The authors should better explain the use of Δcoq9 cells. Similarly, why does truncating the C-terminus of Ptc5 help to discriminate between the unprocessed and mature forms? More explanation of the rationale behind the experiments is needed.

We have now added a sentence on *coq9*. Differences in migration are easier to detect for smaller proteins.

Ptc5 D302/D424 mutants are used in Fig. 3g – I presume these are the phosphatase-dead ones, but this is not mentioned anywhere. Similarly, D84K mutant in Suppl. Fig 8.

We now describe these mutants in the figure legends.

Fig S12. Why does peroxisomal/cytosolic Gpd1 migrate the same as mitochondrial porin in the gradient?

We did not wash the crude organelle fraction. Hence, there was still cytosol in the gradient. Soluble proteins do not entirely run through the gradient but mostly remain in the upper fractions. We now show a gradient, which is based on a two-step purification procedure. The postnuclear supernatant was subjected to an additional 13k spin to remove the cytosolic proteins.

Figure S14 b suggests regulation of the tug-of-war mechanism as one cell shows only mitochondrial targeting and one peroxisome targeting. Is this a one off random event, or observed routinely? Why is Dpi8-RFP-PTS again added here (see Suppl. Figure 13)?

This was routinely observed. Therefore, we thought it is interesting to mention. The picture is added because it nicely shows cells side by side with only mitochondrial and only peroxisomal Dpi8-RFP-PTS, respectively.

The authors address the function of Ptc5 at peroxisomes and nicely show that it dephosphorylates Gpd1 which they suggest increases Gpd1 enzyme activity so is an activator. What does this then mean with respect to cell physiology? When does Gpd1 need to be activated? Can this somehow be linked to IMP1 activation?

Gpd1 is involved in peroxisomal NADH homeostasis. The single deletion only has a mild phenotype on oleic acid. To our knowledge, it is unknown, if Imp1 activity is regulated. It is thought to be a constitutively active protease required for processing of several mitochondrial proteins.

Suppl. Excel sheets: The authors may want to improve annotation, e.g. explain pI4w53. Is this a list of all the potential proteins with a mitochondrial targeting signal at the N-terminus and a C-terminal PTS1 in yeast? What does targeted mean? Have these all been tested? RML2 had a low "score" but was targeted to mitochondria (but not peroxisomes) based on the data presented (?). Please explain.

We improved the table as suggested. We tested all validated mitochondrial proteins (proteomics data; Morgenstern et al. 2017, Table S3) for a PTS1 bioinformatically. Score refers to the probability of sequences harboring a functional PTS1 based on a program published by Neuberger et al. 2003.

REVIEWERS' COMMENTS:

Reviewer #1 (Remarks to the Author):

The revised manuscript by Stehlik et al. includes many important additions that support their original claims. The two biggest concerns of the original manuscript were:

1) Whether Ptc5 truly transited the mitochondria before targeting to the peroxisomes. The original data used to support this claim utilised cleavage of Ptc5 by a mitochondrial protease as a surrogate for mitochondrial transit; however, it was possible that cleavage could also occur in peroxisomes.

2) It was possible that the bulkiness of RFP in the RFP-fusion of Ptc5 led to artefacts in localisation and many of the findings were dependent on this construct.

The inclusion of data within Figures 2-4 of the revised manuscript go a long way to address these concerns. In the case of Point 1) the authors use a Ptc5 construct lacking its mitochondria targeting sequence. As predicted by their model, this construct is completely peroxisomally localised in an uncleaved form. It is perhaps a little surprising that there isn't any difference in the migration of the -MTS construct in comparison to the uncleaved Ptc51-201-RFP-PTS in Figure 4C; however, taking these findings in aggregate with their Imp2 localisation data, their model is convincing. To address the concern with the RFP tag, the authors use an alternative, less bulky tag (3xMyc). This construct convincingly localises to both mitochondria and peroxisomes and therefore supports their original claim that Ptc5 targets to both subcellular compartments. Alongside the data described above, all other technical details and minor points have been addressed.

Reviewer #2 (Remarks to the Author):

All my previous issues have been satisfactorily clarified by the authors. I recommend acceptance of the paper

Reviewer #3 (Remarks to the Author):

This Reviewer thanks the authors for their responses to my queries and concerns.

Reviewer #4 (Remarks to the Author):

In their revised manuscript, the authors have addressed the majority of the comments and suggestions. The whole paper is clearer now, with a better rationale behind the experiments and some data improved. The paper makes a number of interesting observations and comes up with a very intriguing and novel model for protein targeting to peroxisomes via mitochondria, which appears to be convincingly evidenced.

However, I still have some issues with the "tethering data". Looking in detail into contact sites may be beyond the scope of the paper, but in that case, the discussion is too speculative – I would suggest either tuning down the discussion with respect to tethering, or to sufficiently address the tethering aspect and add additional data (see below).

As it stands, the data do not convincingly show that this is a tether. The new data indicate that for the Ptc5-Myc-PTS1 protein only 10% is associated with peroxisomes compared with 30% in the Ptc5-RFP-PTS1 version. Reviewer #2 points out that the RFP moiety in the fusion protein may fold immediately after synthesis in the cytosol leading to an artificial arrest of the protein at the mitochondrial import sites. All the peroxisome-mitochondria association data is based on over-expression of the RFP version. If this RFP causes stalling at the mitochondrial import sites then this

may generate an artificial tether and calls into question this data. At the very least the authors may want to express the HA/myc version and measure peroxisome-mitochondria associations with this construct. Electron microscopy may be optimal to determine peroxisome-mitochondria associations to not rely on fluorescence microscopy. Analysis of the Ptc5 deletion strain may also provide further insight in peroxisome-mitochondria association.

Minor point

I would recommend putting molecular mass markers on the blots in the main Figs. – as the data deal with processing/size changes of bands. I do not find MW markers in the source data as indicated.

First of all the authors want to thank all reviewers for their valuable comments.

Reviewer #1 (Remarks to the Author):

The revised manuscript by Stehlik et al. includes many important additions that support their original claims. The two biggest concerns of the original manuscript were:
1) Whether Ptc5 truly transited the mitochondria before targeting to the peroxisomes. The original data used to support this claim utilised cleavage of Ptc5 by a mitochondrial protease as a surrogate for mitochondrial transit; however, it was possible that cleavage could also occur in peroxisomes.
2) It was possible that the bulkiness of RFP in the RFP-fusion of Ptc5 led to artefacts in localisation and many of the findings were dependent on this construct.
The inclusion of data within Figures 2-4 of the revised manuscript go a long way to address these concerns. In the case of Point 1) the authors use a Ptc5 construct lacking its mitochondria targeting sequence. As predicted by their model, this construct is completely peroxisomally localised in an uncleaved form. It is perhaps a little surprising that there isn't any difference in the migration of the -MTS construct in comparison to the uncleaved Ptc51-201-RFP-PTS in Figure 4C

A size difference for these fusion proteins is not expected. Note that Ptc51-201-RFP-PTS is likely to be cleaved consecutively at two different positions: First, the MTS is cleaved off in the mitochondrial matrix (this is supported by bioinformatic predictions as well as our MS data). The second cleavage reaction catalyzed by the IMP-complex usually leads to the release of soluble proteins into the IMS. In the case of Ptc5 this also allows for translocation into peroxisomes. Since in the *imp1* mutant Ptc51-201-RFP-PTS is likely targeted to the mitochondrial matrix where the MTS is removed the resulting protein should be identical to the -MTS construct.

however, taking these findings in aggregate with their Imp2 localisation data, their model is convincing. To address the concern with the RFP tag, the authors use an alternative, less bulky tag (3xMyc). This construct convincingly localises to both mitochondria and peroxisomes and therefore supports their original claim that Ptc5 targets to both subcellular compartments. Alongside the data described above, all other technical details and minor points have been addressed.

Reviewer #2 (Remarks to the Author):

All my previous issues have been satisfactorily clarified by the authors. I recommend acceptance of the paper

Reviewer #3 (Remarks to the Author):

This Reviewer thanks the authors for their responses to my queries and concerns.

Reviewer #4 (Remarks to the Author):

In their revised manuscript, the authors have addressed the majority of the comments and suggestions. The whole paper is clearer now, with a better rationale behind the experiments and some data improved. The paper makes a number of interesting observations and comes up with a very intriguing and novel model for protein targeting to peroxisomes via mitochondria, which appears to be convincingly evidenced.

However, I still have some issues with the "tethering data". Looking in detail into contact sites may be beyond the scope of the paper, but in that case, the discussion is too speculative – I would suggest either tuning down the discussion with respect to tethering, or to sufficiently address the tethering aspect and add additional data (see below).

As it stands, the data do not convincingly show that this is a tether. The new data indicate that for the Ptc5-Myc-PTS1 protein only 10% is associated with peroxisomes compared with 30% in the Ptc5-RFP-PTS1 version. Reviewer #2 points out that the RFP moiety in the fusion protein may fold immediately after synthesis in the cytosol leading to an artificial arrest of the protein at the mitochondrial import sites. All the peroxisome-mitochondria association data is based on over-expression of the RFP version. If this RFP causes stalling at the mitochondrial import sites then this

may generate an artificial tether and calls into question this data. At the very least the authors may want to express the HA/myc version and measure peroxisome-mitochondria associations with this construct. Electron microscopy may be optimal to determine peroxisome-mitochondria associations to not rely on fluorescence microscopy. Analysis of the Ptc5 deletion strain may also provide further insight in peroxisome-mitochondria association.

The reviewer makes a fair point. Therefore, we toned down the discussion and further clarify that we observe these interactions upon overexpression of RFP-tagged constructs. We also discuss folding of RFP prior to mitochondrial import in the manuscript. However, we also think that due to the domain structure of these proteins with N-terminal mitochondrial presequence and C-terminal PTS1 it is rather difficult to imagine that these molecules do not act as tethers at all, since they will be pulled in opposite directions by the import machineries of both organelles at least to some extent.

Please note that we do not suggest Ptc5 to be a tether as this particular protein is either cleaved by the IMP complex or laterally released into the inner mitochondrial membrane in the absence of IMP. Accordingly, we do not detect more peroxisomes associated with mitochondria upon deletion of *imp1*.

Minor point

I would recommend putting molecular mass markers on the blots in the main Figs. – as the data deal with processing/size changes of bands. I do not find MW markers in the source data as indicated.

The original MW markers were added to all blots showing processing of Ptc5 in the source data file (page ruler prestained). We now labeled the respective blots with molecular weight markers in this file.